# Global ocean methane emissions dominated by shallow coastal waters

Thomas Weber [1*], Nicola A. Wiseman [1,2] & Annette Kock [3]

Oceanic emissions represent a highly uncertain term in the natural atmospheric methane ($CH_4$) budget, due to the sparse sampling of dissolved $CH_4$ in the marine environment. Here we overcome this limitation by training machine-learning models to map the surface distribution of methane disequilibrium ($\Delta CH_4$). Our approach yields a global diffusive $CH_4$ flux of 2–6 Tg $CH_4$ yr$^{-1}$ from the ocean to the atmosphere, after propagating uncertainties in $\Delta CH_4$ and gas transfer velocity. Combined with constraints on bubble-driven ebullitive fluxes, we place total oceanic $CH_4$ emissions between 6–12 Tg $CH_4$ yr$^{-1}$, narrowing the range adopted by recent atmospheric budgets (5–25 Tg $CH_4$ yr$^{-1}$) by a factor of three. The global flux is dominated by shallow near-shore environments, where $CH_4$ released from the seafloor can escape to the atmosphere before oxidation. In the open ocean, our models reveal a significant relationship between $\Delta CH_4$ and primary production that is consistent with hypothesized pathways of in situ methane production during organic matter cycling.

[1] Department of Earth and Environmental Science, University of Rochester, Rochester, NY 14627, USA. [2] Department of Earth System Science, University of California, Irvine, CA 92697, USA. [3] GEOMAR Helmholtz Centre for Ocean Research Kiel, Düsternbrooker Weg 20, 24105 Kiel, Germany. *email: t.weber@rochester.edu

Methane ($CH_4$) is a potent greenhouse gas with a 100-year global warming potential that is ~23 times that of carbon dioxide[1]. Its atmospheric mixing ratio has increased more than two-fold since the preindustrial, contributing ~20% of the radiative climate forcing for all greenhouse gases[2]. Future anthropogenic impacts on the atmospheric $CH_4$ budget are not restricted to direct emissions (e.g. during agriculture and energy production), but will also include climate-driven perturbation of the natural $CH_4$ cycle[3]. This motivates recent efforts to place strong baseline constraints on natural $CH_4$ sources and understand their environmental sensitivity[4].

The global ocean is a highly uncertain term in the atmospheric $CH_4$ budget, emitting 5–25 Tg of $CH_4$ per year (hereafter $Tg\,yr^{-1}$) or 1–13% of all natural emissions[4]. The dominant source of this methane is traditionally thought to be the sea floor, where it is produced biologically in anoxic sediments[5] or released from geological reservoirs at hydrocarbon seeps[6] and degrading methane hydrate deposits[7]. Methane is emitted to the atmosphere by two processes: diffusive gas transfer and ebullition (i.e. bubbling) across the air–sea interface[8]. Ebullitive emissions are only significant in regions that combine very shallow water columns with aggressive rates of $CH_4$ bubbling through the seafloor[9]. Elsewhere, efficient dissolution of $CH_4$ from rising bubbles produces supersaturated waters that drive a diffusive flux to the atmosphere[10], although this pathway is limited by rapid oxidation of dissolved $CH_4$ during its transport through the water column[11]. More recently, novel methanogenesis pathways have been identified that may produce $CH_4$ in situ in the surface ocean mixed layer, providing a more direct conduit to atmosphere[12–14].

Globally, both diffusive and ebullitive $CH_4$ emissions remain uncertain due to sparse data constraints and the crude extrapolation methods used to upscale their rates[4], limiting our understanding of the ocean's leverage over atmospheric $CH_4$. In this study, we provide a new robust estimate for the global diffusive flux and combine it with upper and lower bounds on ebullition rates, thus narrowing the uncertainty range for the total oceanic methane source.

## Results

### Global distribution of methane disequilibrium.
Diffusive air–sea gas fluxes can be estimated from their ocean–atmosphere disequilibrium (denoted $\Delta$) using gas transfer theory[15]. Previous attempts to constrain marine diffusive $CH_4$ emissions have extrapolated from limited cruise track data, estimating a global flux between 0.2 and 18 $Tg\,yr^{-1}$ to the atmosphere[16–19]. We improved upon this approach using machine-learning models to map methane disequilibrium ($\Delta CH_4$) at the global scale, before computing the air–sea flux.

Our work is underpinned by a large compilation of shipboard $CH_4$ concentration measurements collected between 1980 and 2016[20,21], which we combined with atmospheric $pCH_4$ from a global monitoring network to determine $\Delta CH_4$ (see the "Methods" section). Data from the surface mixed layer was then assembled into a monthly climatology at 0.25° horizontal resolution (Fig. 1a, see the "Methods" section). This $\Delta CH_4$ climatology shows that open ocean waters (>2000 m deep) are most weakly supersaturated (0.02–0.2 nM, IQ range), reaching undersaturation in some polar regions (Fig. 1a, b). Surface supersaturation increases sharply towards coastlines, typically ranging between 0.08 and 0.7 nM across continental slopes (200–2000 m), 0.1–2 nM on the outer shelf (50–200 m), and 0.7–20 nM in near-shore environments (0–50 m). In these very shallow waters, $\Delta CH_4$ can occasionally reach many hundreds of nM (~5% above 100 nM, maximum of ~1500 nM). Our climatology contains 8725 gridded data points that are well distributed between marine environments, with ~65% coming from the open ocean and ~10% each from the slope, outer shelf and near-shore regions (Fig. 1b). Normalizing by their areas, this means that data density increases towards coastal waters that are critical regions of elevated flux (Fig. 1b)[22].

Our database is still too sparse for traditional gap-filling approaches applied to oceanographic data (e.g. ref. [23]), especially given the sharp spatial gradients in $\Delta CH_4$. We therefore employed two different machine-learning methods that have previously been applied to map sparse marine data[24–26]: artificial neural networks (ANN) and random regression forests (RRF). These methods build nonlinear statistical models for $\Delta CH_4$ based on its relationship to physical and biogeochemical predictor variables, whose distributions are well known and are plausibly linked to $\Delta CH_4$ (see the "Methods" section), allowing global extrapolation of $\Delta CH_4$ in the mixed layer (Fig. 2a, b). Both ANN and RRF models are trained using randomly selected subsets of the data, and are designed to maximize the prediction of residual validation data while minimizing overfitting (Supplementary Fig. 1). Repeating the training process generates a large ensemble of maps that are used for error propagation (see the "Methods" section).

The machine-learning methods accurately capture the observed magnitude, variance, and spatial patterns of $\Delta CH_4$ both regionally (Supplementary Figs. 2 and 3) and globally (Fig. 2c; $R^2 = 0.7$–0.8 for log-transformed data, see the "Methods" section). They dramatically outperform traditional linear regression ($R^2 = 0$–0.15) and multiple linear regression ($R^2 = 0.2$) models developed from the same predictor variables, according to multiple metrics of model skill (Fig. 2c).

### Diffusive ocean–atmosphere methane flux.
Having mapped $\Delta CH_4$ across the global ocean, we computed the diffusive sea–air $CH_4$ flux at daily resolution using a wind-dependent gas transfer velocity ($k$) and accounting for sea ice cover, which acts as a barrier to gas exchange[27] (see the "Methods" section). A Monte Carlo method was used to propagate uncertainties in $\Delta CH_4$, gas transfer velocity, and ice coverage into our calculation (see the "Methods" section), generating an ensemble of 200,000 different flux estimates (100,000 each for ANN and RRF methods).

The spatial pattern of air–sea flux predicted by these model ensembles is qualitatively similar to the $\Delta CH_4$ distribution, with highest fluxes in shallow shelf regions that often exceed rates of 10 $mmol\,m^{-2}\,yr^{-1}$ (Fig. 3, Supplementary Table 1). Only in outer shelf environments of the Arctic Ocean is there a strong mismatch between the magnitude of $\Delta CH_4$ and flux, due to ice coverage over most of the year. The open ocean is mostly a weak source of $CH_4$ (generally 0–0.5 $mmol\,m^{-2}\,yr^{-1}$), with the exception of the Southern Ocean, which takes up ~0.04 mmol $m^{-2}\,yr^{-1}$ on average south of 45°S. The North Atlantic Ocean polewards of 45°N is either a weak sink (ANN method) or weak source (RRF method) of $CH_4$, marking the only region where the two mapping methods systematically disagree (Figs. 2 and 3), likely due to data scarcity (Fig. 1).

Integrating the fluxes regionally across near-shore, shelf, slope, and open ocean regions reveals a highly disproportionate contribution of shallow waters to oceanic methane emissions (Fig. 4a). The near-shore environment contributes the largest but most uncertain diffusive flux of the four, despite accounting for only ~3% of the ocean area. Emissions in these environments sum to $2.1 \pm 1.6$ and $2.0 \pm 1.45$ $Tg\,yr^{-1}$ (mean ± s.d.) according to the ANN and RRF methods, respectively, with a likely range (defined here as 10–90th percentile range) between 0.8 and 3.8 $Tg\,yr^{-1}$ when ensembles from both mapping methods are combined (Fig. 3a). The open ocean is the second largest emitter (likely

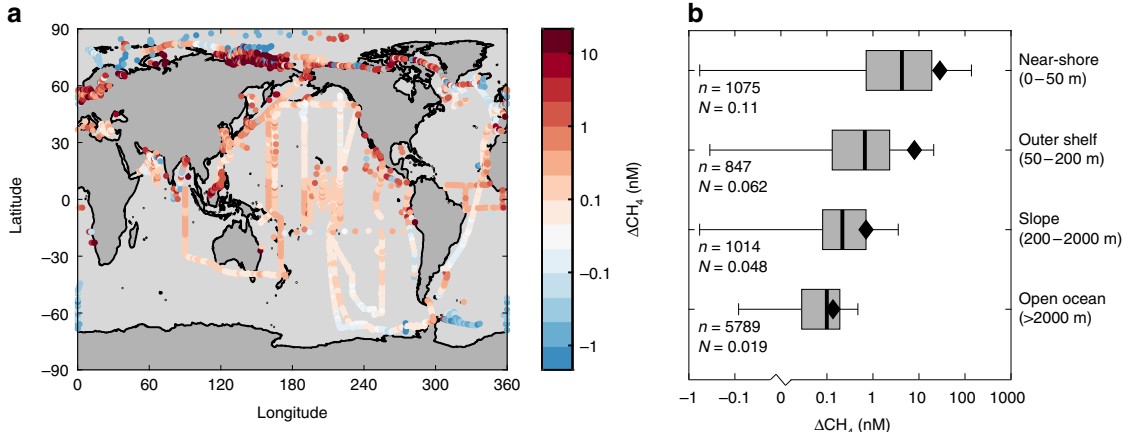

**Fig. 1** Global $\Delta CH_4$ climatology. **a** Annual-mean $\Delta CH_4$, computed after binning all data into 0.25 × 0.25 monthly climatology. Data points are drawn larger than the grid cells for clarity. **b** Probability distributions of observed $\Delta CH_4$, grouped into four bathymetric regions (see also Supplementary Fig. 2). Boxes span the interquartile range, with black line at median. Black diamonds are mean values, and whiskers span the 5–95th percentiles. Number of datapoints ($n$) and data density per $10^9\,m^2$ ($N$) after binning are listed

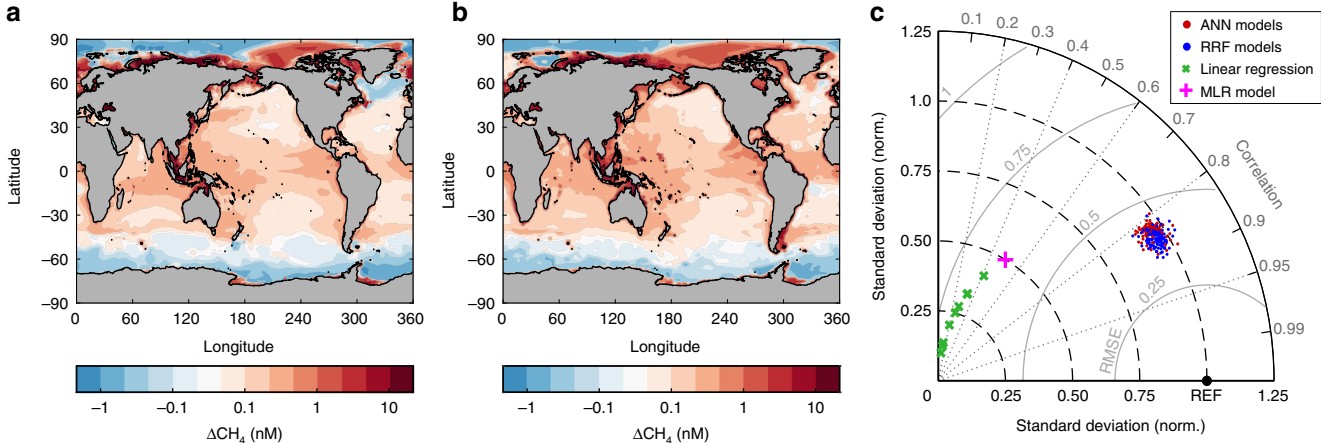

**Fig. 2** Machine-learning mapping of $\Delta CH_4$. **a** Annual mean $\Delta CH_4$ averaged across an ensemble of 100,000 individual maps generated by the artificial neural network (ANN) method. **b** Same as **a** but from random regression forest (RRF) method. **c** Taylor diagram summarizing the fit of a subset of 100 randomly selected ANN and RRF models to observed $\Delta CH_4$, after transformation (see the "Methods" section). Correlation coefficient ($R$) is shown on the outer angular axis, centered root-mean-squared difference is given by radial distance from REF point, and standard deviation (s.d.) normalized by observed s.d. is the radial distance from the origin (points on the 1.0 line have the same s.d. as observations). ANN and RRF dramatically outperform linear regression and multiple linear regression models by all three metrics

range 0.6–1.4 Tg yr$^{-1}$) because its vast area (~85% of ocean) compensates for low flux rates (Fig. 3a), followed by outer shelf (likely range 0.3–1.0 Tg yr$^{-1}$) and continental slope (likely range 0.2–0.6 Tg yr$^{-1}$) environments. Integrated globally, we find an ocean–atmosphere $CH_4$ flux of 4.3 ± 2.2 or 3.9 ± 1.8 Tg yr$^{-1}$ (mean ± s.d.) in the ANN and RRF ensembles, respectively, with a likely range between 2.2 and 6.3 Tg yr$^{-1}$ combining all estimates (Fig. 4b).

Sensitivity tests revealed that the global flux is relatively insensitive to increasing the model grid resolution, the choice of biological predictor variables, and the propagation of potential measurement errors (Supplementary Figs. 1 and 4)[28]. We found that the largest contributor to the range of flux estimates is uncertainty in the $\Delta CH_4$ distribution introduced by our mapping methods, although uncertainty in the gas transfer velocity also makes a significant contribution (Supplementary Figs. 5 and 6).

Our new global estimate of 2.2–6.3 Tg yr$^{-1}$ is larger than previous estimates based on basin-scale cruises (0.2–3 Tg yr$^{-1}$)[16–18], which may have undersampled strongly supersaturated coastal waters, but significantly smaller than estimated from a compilation of shelf data

(11–18 Tg yr$^{-1}$)[19], which likely extrapolated high $\Delta CH_4$ too broadly[18]. In the Arctic Ocean—a region where methane emissions are highly sensitive to future climate warming[29]—we find annual diffusive $CH_4$ emissions of ~0.5 Tg yr$^{-1}$ (Supplementary Table 1). This is substantially lower than a previous estimate from the East Siberian Arctic Shelf[30] (3.3 Tg yr$^{-1}$), despite the fact that our statistical mapping methods skillfully reproduce the $\Delta CH_4$ distribution in this region (Supplementary Fig. 3). This implies that total Arctic $CH_4$ emissions have previously been overestimated, consistent with more recent oceanic and atmospheric observations in this region[31–33].

**Ebullitive and total oceanic methane emissions.** Direct constraints on methane ebullition across the air–sea interface are extremely rare[34], meaning that our statistical mapping methods cannot be applied to scale-up this process. Instead, we attempt to place upper and lower bounds on the global ebullitive emission rate by combining previous estimates of ebullition at the seafloor with bubble model calculations to predict the transfer efficiency of $CH_4$ from the seafloor to the atmosphere.

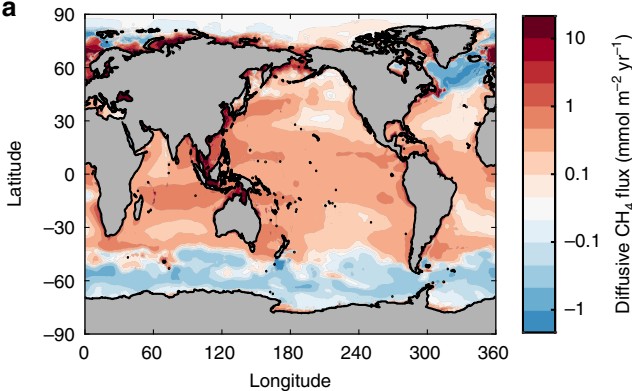

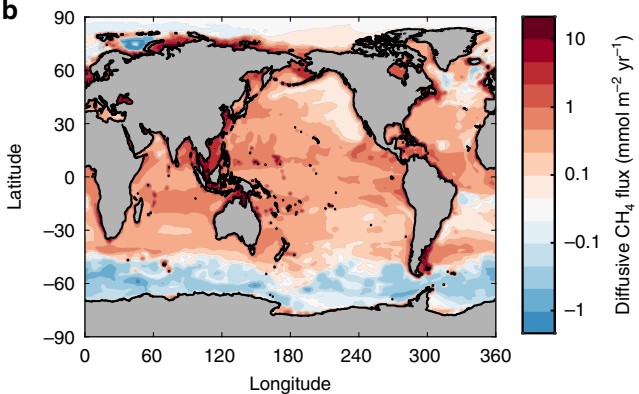

**Fig. 3** Diffusive ocean–atmosphere $CH_4$ flux. **a** Annual diffusive $CH_4$ emissions, averaged across an ensemble 100,000 individual calculations using the artificial neural network mapping method. **b** Same as **a**, but using the random regression forest mapping method

Extrapolation of rate measurements from active seafloor seeps across areas of likely seepage suggests that global $CH_4$ ebullition from continental shelf sediments (0–200 m) likely falls between 18 and 48 Tg yr$^{-1}$ [9,35], with a most likely rate of ~35 Tg yr$^{-1}$ [8,36]. Due to its rapid diffusion from bubbles, the fraction of this $CH_4$ that reaches the atmosphere is governed by the release depth and size-dependent rise velocity of bubbles, and is estimated here using a numerical bubble model that has been validated against observations[10] (see the "Methods" section). Recent observations from high-resolution imaging[37] show that the vast majority of bubbles escaping seafloor sediments (~99% by volume) are between 2 and 8 mm in diameter (Supplementary Fig. 7). Even the largest of these bubbles lose > 99% of their initial $CH_4$ when rising through a 100 m water column (Fig. 5a), suggesting that seeps beyond the continental shelf[7] transfer negligible $CH_4$ to the atmosphere and can be omitted from our global estimate, which is further supported by recent isotopic constraints[38].

Integrated across a representative bubble size spectrum with a volume-weighted mean diameter of ~4 mm (Supplementary Fig. 7)[37], $CH_4$ transfer to the atmosphere decreases rapidly as a function of release depth, even in water columns tens of meters deep (Fig. 5a). The distribution of seeps across the continental shelf is therefore an important determinant of ebullitive emissions, but remains poorly constrained[9]. Based on a compilation of shelf seep locations[35], we consider two limiting scenarios (see the "Methods" section): one in which seeps are uniformly distributed between 0 and 200 m and another in which seeps are confined to waters shallower than 100 m, in which 11% and 17% of the ebullitive $CH_4$ flux is transferred to the atmosphere, respectively (see the "Methods" section).

Applying a transfer efficiency range of 11–17% to seafloor ebullition rates of 35 or 18–48 Tg yr$^{-1}$, we estimate global ebullitive emissions of 4–6 or 2–8 Tg yr$^{-1}$ respectively, which overlap a previous estimate of 0.5–12 Tg yr$^{-1}$ based on simpler bubble transfer assumptions[39,40]. Combined with our probability distributions for diffusive fluxes (Fig. 4b), this implies that the global ocean likely emits 7–11 or 6–12 Tg yr$^{-1}$ of $CH_4$ to the atmosphere (10–90th percentile range, Fig. 5b, see the "Methods" section), depending on the degree of uncertainty in seafloor ebullition rates. Even the broader estimate of 6–12 Tg yr$^{-1}$ constrains oceanic emissions towards the lower end of the range incorporated in previous atmospheric budgets (5–25 Tg yr$^{-1}$) [4]. The previous range incorporates assumptions and extrapolations that have not been updated in many years[41], and can be replaced by our new robust estimate in future appraisals. In part, this will help close the gap between bottom-up estimates of natural $CH_4$ emissions, and the lower rates implied by top-down atmospheric constraints[4].

## Discussion

While our machine-learning models cannot directly constrain the origins of $CH_4$ in the surface ocean, the large-scale distribution of $\Delta CH_4$ they infer may provide useful insights into production mechanisms. We employed a correlation analysis (see the "Methods" section) to determine which of our set of physical and biogeochemical predictor variables most closely approximates the ensemble-mean distribution of $\Delta CH_4$ mapped by our machine learning models (Supplementary Table 2 and Fig. 6).

In coastal ocean regions (<2000 m) where $\Delta CH_4$ spans orders of magnitude, $\log_{10}(\Delta CH_4)$ correlates strongly with seafloor depth ($z_{sf}$, $R^2 = 0.37$), whereas other predictor variables can explain at most ~10% of its spatial variance (Supplementary Table 2). The correlation is further strengthened against $\log_{10}(z_{sf})$ ($R^2 = 0.55$), indicating that the first-order pattern identified by our machine-learning models is a decline in $\Delta CH_4$ away from coastlines following a power-law relationship: $\Delta CH_4 = 67 z_{sf}^{-0.7}$. A similar relationship can be derived directly from the raw dataset used to train our models (Fig. 6b), and the same qualitative pattern is apparent in observations across the shelf at individual locations[22]. The strong dependence of $\Delta CH_4$ on depth reflects the important role of the seafloor as a $CH_4$ source to the surface ocean in coastal regions, supplied by rising gas bubbles that dissolve within meters of the seafloor (Fig. 5a), or by diffusion from anoxic sediments followed by transport to the surface. In the latter case, bathymetry controls both the rain rate of organic carbon that fuels anaerobic metabolism in sediments[8], and the mixing timescale between bottom waters and the surface. The lack of strong relationships with other predictor variables suggests that the environmental controls of seafloor $CH_4$ sources are complex and vary significantly between regions.

Beyond the continental slope (>2000 m), the more subtle open-ocean gradients in $\Delta CH_4$ no longer resemble bathymetry ($R^2 = 2 \times 10^{-5}$), and the almost ubiquitous $CH_4$ supersaturation implies in situ production in the water column rather than transfer from the sediments[8]. Without such a source, rapid $CH_4$ oxidation in the marine environment should leave surface waters undersaturated, driving ingassing from the atmosphere. We only find this condition in the Southern Ocean (Fig. 2), where extensive upwelling supplies $CH_4$-depleted deep water to the surface, and in the central Arctic Ocean, where ice cover mostly prevents air–sea exchange (Supplementary Fig. 4). The predictor variable that most closely approximates ensemble-mean $\Delta CH_4$ in the open ocean is net primary production (NPP), as determined from a carbon-based satellite algorithm[42]. The two are positively correlated and NPP explains ~30% of the variance in $\Delta CH_4$, and ~95%

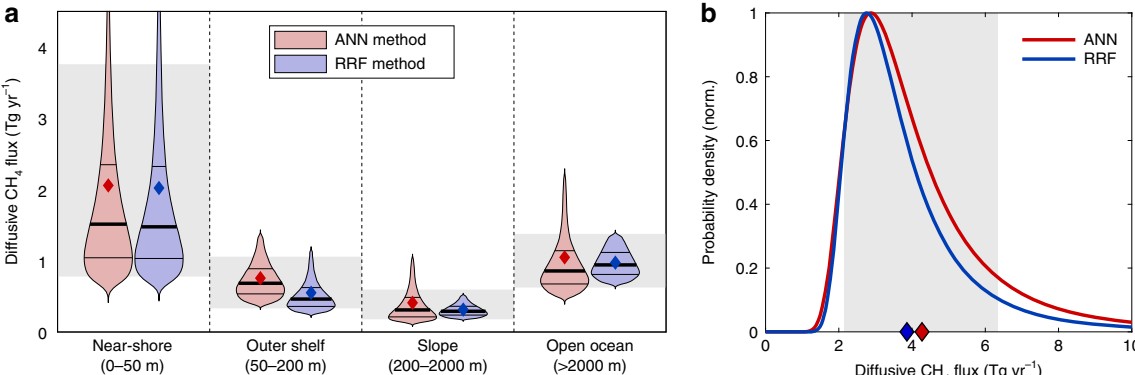

**Fig. 4** Regional and global diffusive $CH_4$ emissions. **a** Violin plot for annual diffusive $CH_4$ emissions integrated across four bathymetric regions, computed using Monte Carlo method to propagate uncertainty in $\Delta CH_4$ and gas transfer velocity. Violin thickness corresponds to probability density, with think black lines at 25th and 75th percentiles, thick line at median, and diamond at mean value. Light gray shading for each region spans the 10–90th percentiles for all estimates, combining artificial network (ANN) and random regression forest (RRF) ensembles. **b** Probability density functions for globally integrated $CH_4$ emissions from ANN and RRF methods. Diamonds and light gray shading as defined in **a**

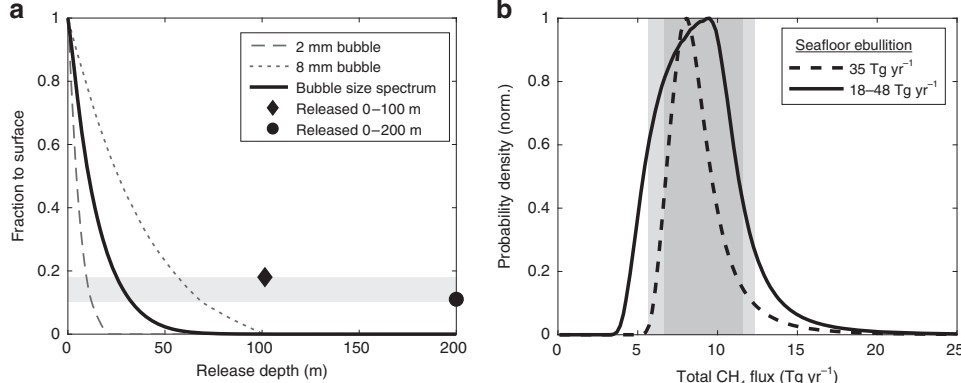

**Fig. 5** Ebullitive and total $CH_4$ emissions. **a** Modeled transfer efficiency of $CH_4$ in bubbles from the seafloor to surface ocean, for 2 and 8 mm diameter bubbles, and integrated across a characteristic bubble size spectrum (Supplementary Fig. 7). Diamond and circle points represent the mean transfer efficiency for bubbles released uniformly between 0–100 and 0–200 m, respectively, and gray shading marks the range of 11–17% bounded by these cases. **b** Probability density functions for total oceanic $CH_4$ emissions, combining the distribution for diffusive fluxes (Fig. 4b) with two uniform probability distributions for ebullitive emissions that are obtained by applying 11–17% transfer efficiency to seafloor ebullition rates of 35 and 18–48 Tg yr$^{-1}$. Dark and light gray shading mark the likely range (10–90th percentiles) for the two estimates

of its large-scale latitudinal pattern, which is highest in the tropics and lowest in polar oceans, with subtropical and subpolar regions falling between (Fig. 6c). A similar although somewhat weaker correlation to NPP emerges from our raw $\Delta CH_4$ database (Fig. 6d), demonstrating that this relationship is not generated artificially during the mapping procedure.

Methane production has been reported during growth of coccolithophores[13] and other ubiquitous members of the prymnesiophyte class of marine phytoplankton[43], which may contribute in part to the correlation we find between $\Delta CH_4$ and NPP. However, a number of alternative pathways have been proposed for methanogenesis in surface ocean waters, which could give rise to the relationship indirectly. $CH_4$ may be released from sinking organic aggregates that harbor anoxic microzones suitable for methanogensis[44], but this should result in a stronger relationship of $\Delta CH_4$ to particulate organic carbon (POC) flux than to NPP, which is not borne out in our analysis ($R^2 = 0.14$, Supplementary Table 2). Similarly, $CH_4$ may be produced in the anoxic digestive tracts of zooplankton and egested to the watercolumn at potentially significant rates[14]. Because zooplankton biomass and productivity scales with NPP[45], this mechanism is broadly consistent with the surface distribution of $\Delta CH_4$.

In addition, two aerobic pathways have been identified for methanogenesis during the microbial cycling of dissolved organic matter (DOM) compounds, which are ultimately a product of phytoplankton growth (i.e. NPP). First, microbial transformations of dimethylsulfide (DMS) are thought to yield $CH_4$ (ref. [46]), but we find only a weak correlation between DMS and $\Delta CH_4$ (Supplementary Table 2), suggesting this is not an important pathway at the global scale. Second, $CH_4$ is produced by the degradation of methylphosphonate[12] (MPn)—an important constituent of the surface DOM inventory[47]—especially under phosphate ($PO_4$) limited conditions. We find that a multiple linear regression model combining a positive relationship to NPP and a negative relationship to $[PO_4]$ explains surface $\Delta CH_4$ significantly better than NPP alone ($\Delta CH_4 = 5 \times 10^{-3}$ NPP–0.1 $[PO_4]$–0.03, $R^2 = 0.35$). This relationship is consistent with timeseries evidence for coincident variations in $\Delta CH_4$ and $[PO_4]$ in the North Pacific Ocean while NPP remained constant[48], and supports an important role for MPn cycling as a $CH_4$ source.

Ultimately, a combination of pathways may control the open ocean surface $\Delta CH_4$ distribution and contribute to its correlation with NPP. Methanogenesis by phytoplankton and in zooplankton guts may dominate in productive ocean regions, with MPn

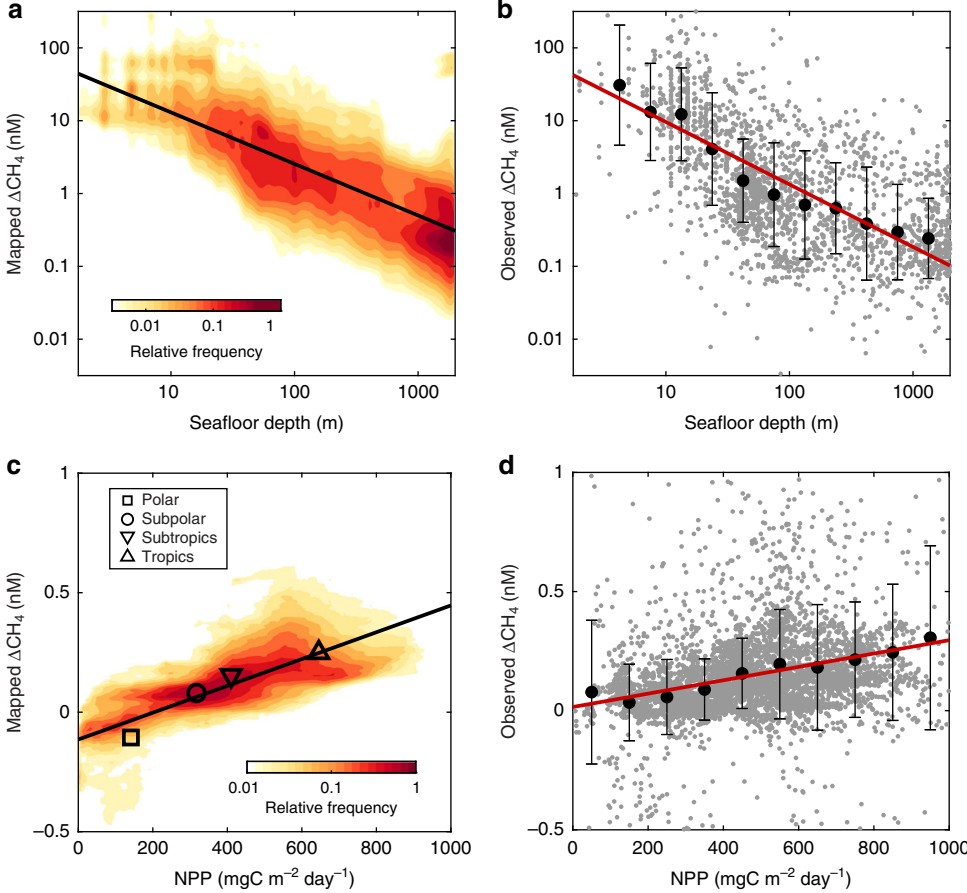

**Fig. 6** Controls on surface ocean $\Delta CH_4$. **a** Joint probability distribution for mapped $\Delta CH_4$ and seafloor depth ($z_{sf}$) in coastal ocean regions (<2000 m depth). Color scale represents the frequency of gridcells with a given combination of $\log_{10}$(depth) and $\log_{10}(\Delta CH_4)$, after averaging together all 200,000 machine-learning maps. Black line is the best fit for the mapped data ($\Delta CH_4 = 67z_{sf}^{-0.7}$, $R^2 = 0.55$). **b** Scatter plot of observed $\Delta CH_4$ versus depth. Gray points show raw data; black circles with errorbars show mean ± s.d. $\Delta CH_4$ within depth bins. Red line is best fit to the binned data ($CH_4 = 69z_{sf}^{-0.8}$, $R^2 = 0.94$). **c**, **d** Same as **a** and **b**, but for the relationship of mapped **c** and observed **d** $\Delta CH_4$ to net primary production (NPP) in open ocean (>2000 m depth) environments. In **c**, black line is the best fit for mapped data ($\Delta CH_4 = (0.5NPP - 62)/10^3$, $R^2 = 0.30$), and symbols represent large-scale averages (Supplementary Fig. S8). In **d**, black circles show mean ± s.d. $\Delta CH_4$ within NPP bins, and red line is best fit to the binned data ($\Delta CH_4 = (0.3NPP + 14)/10^3$, $R^2 = 0.91$)

becoming the dominant pathway in oligotrophic regions, where $PO_4$ stress acts as the driving variable by selecting for phosphonate decomposing metabolisms[49]. Additionally, we cannot definitively conclude that the NPP vs. $CH_4$ relationship arises mechanistically from methanogenesis, and not from spatial variations in $CH_4$ oxidation or the physical $CH_4$ supply, which may also be correlated with NPP.

This work has narrowed the uncertainty range of total oceanic $CH_4$ emissions to 6–12 Tg yr$^{-1}$, providing a robust baseline to assess anthropogenic perturbations against, and contributing towards an improved accounting of the natural atmospheric methane budget. The majority of the remaining uncertainty in our estimate is attributed to shallow near-shore environments, where $\Delta CH_4$ and diffusive emissions vary most among our model ensembles (Fig. 4a), and where relatively unconstrained ebullitive fluxes are concentrated (Fig. 5a). To further refine our estimate, future observational efforts should focus on these shallow environments and sample with the resolution to capture sharp coastal gradients in $\Delta CH_4$ (ref. [22]), while employing new imaging technologies[37] to further constrain bubble dynamics and ebullition. Understanding and resolving interlaboratory discrepancies in [$CH_4$] measurements[28] should also be prioritized, so that consistent data may be synthesized across multiple sources.

By contrast, open ocean $CH_4$ emissions are relatively well constrained (Fig. 4a) and are driven by $\Delta CH_4$ variations that appear systematically linked to organic matter cycling (Fig. 6). Our work supports previous hypotheses for $CH_4$ release during phytoplankton growth, zooplankton egestion, and MPn degradation, and we encourage future work to distinguish and quantify the contributions of these process. The global relationship between $\Delta CH_4$ and NPP reported here also potentially provides a simple approach to represent open ocean emissions in coupled ocean–atmosphere models, and tentatively predict future perturbations in this source as ocean warming and stratification impact marine productivity[50].

## Methods

**$CH_4$ concentration database**. We compiled a large database of $CH_4$ concentration measurements from the ocean mixed layer, to form the basis of a $\Delta CH_4$ climatology that was used train machine-learning models. The majority of [$CH_4$] data were taken from the MarinE MethanE and NiTrous Oxide (MEMENTO) Database, which has compiled published trace gas measurements from research cruises dating back to 1970 (ref. [20,21]). The full dataset and references for individual data contributions can be found at https://memento.geomar.de. We downloaded the version of MEMENTO available as of June 2018, and retained only data that was collected within the mixed layer depth, as determined by interpolation from the MIMOC global mixed layer climatology[51]. We rejected data points that were not accompanied by temperature data, which is required to compute $CH_4$ solubility. Data points with missing salinity data were accepted, due to its weaker effect on

solubility, and salinity was filled by interpolating from the MIMOC salinity climatology[51]. We also rejected data collected outside the time interval 1980–2016, when atmospheric $pCH_4$ could not be determined (see below).

We combined this subset of the MEMENTO database with other recent published $[CH_4]$ measurements from the surface ocean to expand data coverage in critical regions, mostly polar oceans and marginal seas[30,32,52–57]. Again, data collected below the climatological mixed layer was rejected, and missing salinity data was filled from MIMOC. Only the data from ref. [30] was accepted without accompanying temperature data, which was filled by interpolation from MIMOC. This data has previously been used to infer very large $CH_4$ emissions from Arctic shelves and was included in our database to test this inference.

**Mixed layer $\Delta CH_4$ climatology.** Each mixed layer $[CH_4]$ measurement in our database was converted to $CH_4$ disequilibrium ($\Delta CH_4$) using:

$$\Delta CH_4 = [CH_4] - S_{CH_4} p_{CH_4}^{moist} \qquad (1)$$

In Eq. (1), $S_{CH_4}$ is the solubility of methane computed from temperature and salinity at each data point[58], and $p_{CH_4}^{moist}$ is the partial pressure of $CH_4$ in moist air. $p_{CH_4}^{moist}$ was determined by first interpolating dry-air $p_{CH_4}$ to the location of each ocean data point from atmospheric measurements taken in the same year and month, using ordinary kriging. Atmospheric data was taken from the NOAA Global Monitoring Division archive, which has collected flask samples from a global network of monitoring stations since 1980 (https://www.esrl.noaa.gov/gmd/ccgg/). Dry $p_{CH_4}$ was then converted to $p_{CH_4}^{moist}$ following ref. [59].

Finally, our complete $\Delta CH_4$ database of ~120,000 observations was compiled into a monthly climatology. For each month, all data collected during that month (regardless of year) was binned onto a $0.25° \times 0.25°$ latitude/longitude grid, and the average value for each grid cell was calculated. This step was necessary to minimize the impact of a few high-resolution cruise tracks, which contribute orders of magnitude more datapoints than others. We note that by combining data from the years 1985–2016 into a single monthly climatology, we have made the implicit assumption that $\Delta CH_4$ remains relatively constant over time, even as atmospheric $p_{CH_4}$ has increased by ~10% from ~1650 to ~1850 ppb. This assumption is supported by observations from open ocean waters in the Atlantic[18,60] and Pacific[17] oceans, where $\Delta CH_4$, and therefore air–sea flux, remained constant over interannual to decadal timescales while $[CH_4]$ increased in track with $p_{CH_4}$. It is consistent with the view that $\Delta CH_4$ is controlled by internal sources and sinks of $CH_4$ that maintain a disequilibrium between the ocean and atmosphere, regardless of the atmospheric mixing ratio[8,18].

**Machine-learning mapping.** Our monthly $\Delta CH_4$ climatology was used to train an ensemble of ANN and RRF models to generate continuous, mapped climatologies. These are both machine-learning methods that exploit pattern similarities between $\Delta CH_4$ and other physical, chemical, and biological properties (termed predictor variables) whose climatological distributions are well known, to generate skillful predictive models for $\Delta CH_4$. Employing the two independent mapping methods and taking an ensemble approach allows us to propagate uncertainties introduced by the mapping process into our flux estimates.

Predictor data used in our models include: seafloor depth taken from the ETOPO2 high-resolution bathymetry (https://rda.ucar.edu/datasets/ds759.3/, available at 0.033° resolution); surface temperature and salinity from the MIMOC climatology (0.5° resolution)[51]; a net primary production (NPP) climatology constructed from data collected between 2002 and 2016 by a carbon-based remote-sensing algorithm (http://www.science.oregonstate.edu/ocean.productivity/, 0.25° resolution); POC export flux at the base of the euphotic zone, estimated by combining our NPP climatology with the export ratio algorithm of ref. [61] phosphate ($[PO_4]$) in the surface ocean, from the World Ocean Atlas 2013 (WOA13) climatology[23] (0.25° resolution); oxygen ($[O_2]$) in shallow subsurface waters (50 m below mixed layer, or at seafloor depth if seafloor is within 50 m of mixed layer) from the WOA13 climatology; sediment gas hydrate inventory, taken from the global model of ref. [62] (1° resolution). All predictor data were interpolated from their original grids to the same $0.25° \times 0.25°$ as the $\Delta CH_4$ climatology. We note that while we have chosen the most up-to-date global data products for use in our work, each is likely subject to its own uncertainties, and some have been subjected to their own gap-filling procedures.

Each ANN and RRF ensemble member was trained using a random subset of 70% of the dataset, leaving 30% of the data for validation. Before training, $\Delta CH_4$ was transformed using an inverse hyperbolic sine (IHS) transform, which is similar to a log transform except it is defined at negative $\Delta CH_4$. Because $\Delta CH_4$ spans more than four orders of magnitude, this transform prevents a few data points with very high $\Delta CH_4$ from dominating the training process. While the transformation is not necessary for the RRF method, it was undertaken for operational consistency between our two approaches.

Our ANN model structure is similar to that used in ref. [25], with a single hidden layer of 20 neurons (sigmoid response functions), fully connected to a single-node output layer (linear response function), and is trained using a Bayesian regularization method. The individual regression trees comprising our RRF ensemble are structured with a maximum of 100 decision splits and trained using a standard CART algorithm. The complexity of these models is chosen to maximize

predictive skill while minimizing overfitting. More complex models (i.e. more neurons in the ANN or more decision splits in RRF trees) achieves a better fit to the full dataset, because the majority of that data is used in training the model. However, when the fit to validation data does not improve in tandem, it suggests the model is overfitting the training data, rather than improving its predictive power. We therefore experimented with different levels of complexity (Supplementary Fig. 1a, b), and chose the level at which the fit to validation data began to plateau.

An ensemble of 100,000 ANN and RRF models was trained for error propagation (see below). All ensemble members were able to reproduce the IHS-transformed validation data with $R > 0.75$, and closely matched the variance of the data (Fig. 2c) and its probability distribution in different environments (Supplementary Fig. 2). After training, each ensemble member was used to generate a $0.25° \times 0.25°$ monthly mapped $\Delta CH_4$ climatology by applying the model to gridded climatologies of the predictor data.

**Diffusive $CH_4$ fluxes and error propagation.** To estimate diffusive $CH_4$ fluxes ($F_{diff}$) across the air–sea interface, we applied a standard gas transfer model to our $\Delta CH_4$ climatologies:

$$F_{diff} = (1 - \varepsilon_{ice} f_{ice}) k \Delta CH_4 \qquad (2)$$

Here, $f_{ice}$ is the fractional sea ice cover of a grid cell, $\varepsilon_{ice}$ is the efficiency with which ice cover blocks gas exchange (1 means no exchange through ice), and $k$ is the gas transfer velocity. A number of different empirical algorithms have been proposed relating $k$ to wind speed at 10 m above the air–sea interface, and diverge by >20% at characteristic ocean wind speeds between 5 and 10 m s$^{-1}$ (ref. [63]). Additionally, a number of wind speed and ice coverage climatologies have been assembled from different methodologies, which all agree in their large-scale patterns but can differ at smaller scales.

To propagate these sources of uncertainty into our flux calculation, we used a Monte Carlo procedure in which each $\Delta CH_4$ climatology was combined in Eq. (2) with random selections between five different wind climatologies, three different sea ice climatologies, and four different empirical algorithms for $k$ (refs. [15,64–66]). We note that the most recent and perhaps best constrained of these algorithms[15] yields $k$ values close to the average of all four. Daily wind climatologies were obtained from the cross-calibrated multi-platform (CCMP) product[67] (http://www.remss.com/measurements/ccmp/) that combines satellite and buoy data with model predictions, the QuickScat product (http://www.remss.com/missions/qscat/) from satellite scatterometry, the WindSat product (http://www.remss.com/missions/windsat/) from satellite radiometry, the ECMWF ERA-Interim product from model reanalysis (https://www.ecmwf.int/en/forecasts/datasets/reanalysis-datasets/era-interim) and the NCEP product (https://www.esrl.noaa.gov/psd/data/gridded/data.ncep.reanalysis.html) from model reanalysis. Monthly sea ice climatologies were obtained from the ECMWF ERA-Interim and NCEP reanalysis products (links above), and the HadISST product that combines in situ and satellite observations (https://catalogue.ceda.ac.uk/uuid/facafa2ae494597166217a9121a62d3c).

Flux calculations were conducted at daily resolution to limit the impact of temporal smoothing of windspeeds, given that the relationship between $k$ and windspeed is nonlinear. Windspeed and $f_{ice}$ climatologies were first interpolated to our $0.25° \times 0.25°$ grid, and then monthly $\Delta CH_4$ and $f_{ice}$ were interpolated to each day of the year before applying Eq. (2). While most estimates of air–sea gas exchange assume that ice coverage completely blocks gas exchange ($\varepsilon_{ice} = 1$), we allow gas transfer across sea ice to occur up to 10% as fast as in ice-free water, based on radon measurements in Arctic Ocean[27]. Each Monte Carlo iteration therefore randomly selected from the range $0.9 < \varepsilon_{ice} < 1$ for application in Eq. (2).

**Sensitivity tests.** To inform our selection of grid resolution, we applied the full procedure outlined above using grids ranging from 2° to 0.125° in resolution (Supplementary Fig. 1). In each case, $\Delta CH_4$ data were binned into a climatology at the specified resolution, predictor variables were interpolated to the specified resolution, and an ensemble of 200 $\Delta CH_4$ and flux estimates were generated (100 each from ANN and RRF). The total global flux decreased as the grid resolution was improved, because coarser grids spread high coastal $\Delta CH_4$ values over larger areas. This trend plateaued between 0.5° and 0.25° resolution, so we selected a 0.25° grid (~25 × 25 km near equator) for our full model ensemble, to balance accuracy and computational efficiency.

To test whether selecting different biological predictor variables would impact our results, we conducted a sensitivity test in which NPP was replaced with the high-resolution MODIS chlorophyll-a (Chl) climatology (https://oceancolor.gsfc.nasa.gov/, 4 km resolution) and a new suite of 200 flux estimates was generated. The global fluxes predicted by this ensemble were not significantly different from those using NPP as the biological predictor variable. Furthermore, improving the grid resolution beyond 0.25° again had no impact on the global flux, suggesting this plateau is not dependent on predictor resolution.

We tested whether potential errors in our $[CH_4]$ database would greatly impact our results, because recent work has revealed interlaboratory discrepancies in $[CH_4]$ measurements[28]. Prior to generating our $\Delta CH_4$ climatology and applying our mapping methods, a synthetic database was generated by randomly selecting a $[CH_4]$ value for each datapoint in the range $(1 - R.E.)[CH_4]_{obs}$ to $(1 + R.E.)$ $[CH_4]_{obs}$, where $[CH_4]_{obs}$ is the reported value. Measurements from individual

laboratories can diverge up to 25% from the interlaboratory mean in strongly supersaturated waters and up to 50% in weakly supersaturated waters[28]. We therefore conducted tests with R.E. = 0.25 and R.E. = 0.5, and generated an ensemble of 200 flux estimates in each case (Supplementary Fig. 4). We find that propagating potential measurement errors does not change the ensemble-mean global diffusive flux (~4 Tg yr$^{-1}$ in each case), but expands the likely range to 1.8–6.4 or 1.5–6.9 Tg yr$^{-1}$ (R.E. = 0.25, 0.5 respectively).

Finally, we attempted to compare the degree of uncertainty introduced to our flux calculations by the $\Delta CH_4$ distribution and by gas transfer velocity (Supplementary Fig. 6). First, each permutation of windspeed climatologies, $f_{ice}$ climatologies, and algorithms for $k$ (60 permutations) was used to calculate $(1 - \varepsilon_{ice} f_{ice})k$, and each was applied to the same $\Delta CH_4$ climatology (average from our full ensemble). Second, the same $(1 - \varepsilon_{ice} f_{ice})k$ climatology (average across the 60 permutations) was applied to 60 different $\Delta CH_4$ maps generated by the ANN and RRF methods. The variance across these two ensembles can be used to compare the uncertainty introduced by gas transfer velocity (first ensemble) versus $\Delta CH_4$ (second ensemble).

**Ebullitive and total $CH_4$ fluxes.** We attempted to place broad bounds on ebullitive $CH_4$ emissions from the ocean. The globally integrated ebullitive flux to the atmosphere ($\Sigma F_{eb}$) can be estimated from:

$$\Sigma F_{eb} = \overline{\varepsilon_{tr}} \Sigma F_{sf} \qquad (3)$$

In Eq. (3), $\Sigma F_{sf}$ is the globally integrated ebullitive flux from the seafloor to the water column, $\varepsilon_{tr}$ denotes the transfer efficiency of the $CH_4$ through the water column and to the atmosphere, and $\overline{\varepsilon_{tr}}$ represents the flux-weighted global average of $\varepsilon_{tr}$. We take two previous literature values of $\Sigma F_{sf}$: the most likely flux of 35 Tg yr$^{-1}$ from ref. [36] and the full range of 18–48 Tg yr$^{-1}$ based on a compilation of seepage rates by ref. [9]. We note that these $\Sigma F_{sf}$ estimates apply only to shelf regions between 0 and 200 m, but because $\varepsilon_{tr}$ approaches 0 in waters beyond the shelf[10,38], this is sufficient to estimate $\Sigma F_{eb}$ (flux to atmosphere).

We estimated $\varepsilon_{tr}$ using output from a model of rising gas bubbles, which simulates the diffusive loss of $CH_4$ to predict the fraction that reaches the surface as a function of bubble size and release depth[10]. Because environmental conditions have a relatively small impact on $CH_4$ transfer in this model, we use model output generated previously under idealized conditions that is recommended for application in most marine environments[10]. First, we integrated this output across a characteristic volume-weighted bubble size distribution to determine $\varepsilon_{tr}$ as a function of release depth (Fig. 5a). This size distribution is generated by combining the individual distributions from four seep sites observed recently using high-resolution imaging[37]. While we note that these observations are from deeper seeps than the shelf seeps we are interested in, the bubble sizes reported are consistent with older, less well-resolved observations from shelf seeps and shallow lake[9,68].

To determine $\overline{\varepsilon_{tr}}$ we must know the depth distribution of the seafloor ebullitive flux ($\Sigma F_{sf}$). While relatively few individual seep locations have been charted, these are widely distributed across continental shelves at depths between 0 and 200 m (ref. [35]). However, some of the world's most active seep sites are situated in waters shallower than 100 m (e.g. Santa Monica Channel, ~60 m; Norwegian North Sea, 60–80 m). Based on these observations, we use two limiting scenarios to bracket $\overline{\varepsilon_{tr}}$. First, to derive a lower limit, we assume that $\Sigma F_{sf}$ is uniformly distributed between 0 and 200 m, and average the depth-dependent $\varepsilon_{tr}$ across this interval, weighted by the ocean area with each depth, yielding $\overline{\varepsilon_{tr}} = 11\%$. To derive an upper limit, we assume that $\Sigma F_{sf}$ is confined to regions between 0 and 100 m depth, and repeat the calculation to yield $\overline{\varepsilon_{tr}} = 17\%$. This range of $\overline{\varepsilon_{tr}}$ (11–17%) was combined with the two estimates of $\Sigma F_{sf}$ (35 and 18–48 Tg yr$^{-1}$) in Eq. (3) to specify likely ranges for $\Sigma F_{eb}$, and we assumed uniform probability within these ranges. Finally, total oceanic $CH_4$ emissions were estimated by combining these uniform probability distributions for $\Sigma F_{eb}$ with the probability distributions derived previously for diffusive fluxes (Fig. 5b).

**Analysis of $\Delta CH_4$ distribution.** To evaluate which physical or biogeochemical properties drive the global distribution of methane disequilibrium in our machine-learning models, we correlated annual-mean mapped $\Delta CH_4$ (averaged across all 200,000 climatologies, Supplementary Fig. 8) against each predictor variable in turn. A climatology of DMS[69] was also correlated against $\Delta CH_4$ to test hypothe-sized production during DMS cycling[46]. This analysis was conducted separately for coastal oceans (<2000 m depth) and the open ocean (>2000 m depth), given that different drivers are likely dominant in these environments[8].

To compare the large-scale open-ocean patterns of NPP and $\Delta CH_4$ across latitude, both variables were averaged across polar, subpolar, subtropical, and tropical regions. In the Southern, Atlantic, and Pacific and Arctic Oceans, these regions were defined as in ref. [70], and the Indian Ocean was split into tropics and subtropics along 15°S (Supplementary Fig. 8).

## Data availability

All datasets used in this work are described in the Methods section, and links are provided to the online repositories where they can be obtained. Gridded climatologies of methane disequilibrium and air-sea methane fluxes generated by this study are available at https://figshare.com/articles/ocean_ch4_nc/9034451.

## Code availability

This work makes extensive use of functions from MATLAB's Machine Learning Toolbox (https://www.mathworks.com/solutions/machine-learning.html). Custom codes for applying those functions are available upon request from the corresponding author.

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

# ARTICLE

25. Roshan, S. & DeVries, T. Efficient dissolved organic carbon production and export in the oligotrophic ocean. *Nat. Commun.* 8, https://doi.org/ARTN 203610.1038/s41467-017-02227-3 (2017).

26. Sherwen, T. et al. A machine learning based global sea-surface iodide distribution. *Earth Syst. Sci. Data Discuss.* **2019**, 1–40 (2019).

27. Rutgers van der Loeff, M. M., Cassar, N., Nicolaus, M., Rabe, B. & Stimac, I. The influence of sea ice cover on air–sea gas exchange estimated with radon-222 profiles. *J. Geophys. Res.-Oceans* 119, 2735–2751 (2014).

28. Wilson, S. T. et al. An intercomparison of oceanic methane and nitrous oxide measurements. *Biogeosciences* 15, 5891–5907 (2018).

29. Biastoch, A. et al. Rising Arctic Ocean temperatures cause gas hydrate destabilization and ocean acidification. *Geophys. Res. Lett.* 38, https://doi.org/10.1029/2011gl047222 (2011).

30. Shakhova, N. et al. Extensive methane venting to the atmosphere from sediments of the East Siberian Arctic Shelf. *Science* 327, 1246–1250 (2010).

31. Berchet, A. et al. Atmospheric constraints on the methane emissions from the East Siberian Shelf. *Atmos. Chem. Phys.* 16, 4147–4157 (2016).

32. Thornton, B. F., Geibel, M. C., Crill, P. M., Humborg, C. & Morth, C. M. Methane fluxes from the sea to the atmosphere across the Siberian shelf seas. *Geophys. Res. Lett.* 43, 5869–5877 (2016).

33. Warwick, N. J. et al. Using δ13C-CH4 and δD-CH4 to constrain Arctic methane emissions. *Atmos. Chem. Phys.* 16, 14891–14908 (2016).

34. Shakhova, N. et al. Ebullition and storm-induced methane release from the East Siberian Arctic Shelf. *Nat. Geosci.* 7, 64, https://doi.org/10.1038/ngeo2007 (2013).

35. Hovland, M., Judd, A. G. & Burke, R. A. The global flux of methane from shallow submarine sediments. *Chemosphere* 26, 559–578 (1993).

36. Kvenvolden, K. A. & Rogers, B. W. Gaia's breath—global methane exhalations. *Mar. Pet. Geol.* 22, 579–590 (2005).

37. Wang, B., Socolofsky, S. A., Breier, J. A. & Seewald, J. S. Observations of bubbles in natural seep flares at MC 118 and GC 600 using in situ quantitative imaging. *J. Geophys. Res.-Oceans* 121, 2203–2230 (2016).

38. Leonte, M. et al. Using carbon isotope fractionation to constrain the extent of methane dissolution into the water column surrounding a natural hydrocarbon gas seep in the Northern Gulf of Mexico. *Geochem. Geophys. Geosyst.* 19, 4459–4475 (2018).

39. Judd, A. et al. Contributions to atmospheric methane by natural seepages on the UK continental shelf. *Mar. Geol.* 137, 165–189 (1997).

40. Judd, A. G. in *Atmospheric Methane: Its Role in the Global Environment* (ed. Mohammad Aslam Khan Khalil) 280–303 (Springer, Berlin, Heidelberg, 2000).

41. Ruppel, C. D. & Kessler, J. D. The interaction of climate change and methane hydrates. *Rev. Geophys.* 55, 126–168 (2017).

42. Behrenfeld, M. J., Boss, E., Siegel, D. A. & Shea, D. M. Carbon-based ocean productivity and phytoplankton physiology from space. *Glob. Biogeochem. Cycles* 19, 3GB1006 (2005).

43. Klintzsch, T. et al. Methane production by three widespread marine phytoplankton species: release rates, precursor compounds, and relevance for the environment. *Biogeosci. Discuss.* **2019**, 1–25 (2019).

44. Sasakawa, M. et al. Carbon isotopic characterization for the origin of excess methane in subsurface seawater. *J. Geophys. Res.* 113, https://doi.org/10.1029/2007jc004217 (2008).

45. Strömberg, K. H. P., Smyth, T. J., Allen, J. I., Pitois, S. & O'Brien, T. D. Estimation of global zooplankton biomass from satellite ocean colour. *J. Mar. Syst.* 78, 18–27 (2009).

46. Florez-Leiva, L., Damm, E. & Farías, L. Methane production induced by dimethylsulfide in surface water of an upwelling ecosystem. *Prog. Oceanogr.* **112–113**, 38–48 (2013).

47. Repeta, D. J. et al. Marine methane paradox explained by bacterial degradation of dissolved organic matter. *Nature Geoscience*, 9, https://doi.org/10.1038/NGEO2837 (2016).

48. Wilson, S. T., Ferrón, S. & Karl, D. M. Interannual variability of methane and nitrous oxide in the North Pacific Subtropical Gyre. *Geophys. Res. Lett.* 44, 9885–9892 (2017).

49. Sosa, O. A., Repeta, D. J., DeLong, E. F., Ashkezari, M. D. & Karl, D. M. Phosphate-limited ocean regions select for bacterial populations enriched in the carbon–phosphorus lyase pathway for phosphonate degradation. *Environ. Microbiol.* 21, 2402–2414 (2019).

50. Bopp, L. et al. Multiple stressors of ocean ecosystems in the 21st century: projections with CMIP5 models. *Biogeosciences* 10, 6225–6245 (2013).

51. Schmidtko, S., Johnson, G. C. & Lyman, J. M. MIMOC: a global monthly isopycnal upper-ocean climatology with mixed layers. *J. Geophys. Res.-Oceans* 118, 1658–1672 (2013).

52. Geprägs, P. et al. Carbon cycling fed by methane seepage at the shallow Cumberland Bay, South Georgia, sub-Antarctic. *Geochem. Geophys. Geosyst.* 17, 1401–1418 (2016).

53. Mau, S. et al. Seasonal methane accumulation and release from a gas emission site in the central North Sea. *Biogeosciences* 12, 5261–5276 (2015).

54. Mau, S. et al. Widespread methane seepage along the continental margin off Svalbard— from Bjørnøya to Kongsfjorden. *Sci. Rep.-UK* 7, 42997 (2017).

55. Steinle, L. et al. Effects of low oxygen concentrations on aerobic methane oxidation in seasonally hypoxic coastal waters. *Biogeosciences* 14, 1631–1645 (2017).

56. Kudo, K. et al. Spatial distribution of dissolved methane and its source in the western Arctic Ocean. *J. Oceanogr.* 74, 305–317 (2018).

57. Yoshikawa, C. et al. Methane sources and sinks in the subtropical South Pacific along 17°S as traced by stable isotope ratios. *Chem. Geol.* 382, 24–31 (2014).

58. Wiesenburg, D. A. & Guinasso, N. L. Jr Equilibrium solubilities of methane, carbon monoxide, and hydrogen in water and sea water. *J. Chem. Eng. Data* 24, 356–360 (1979).

59. Weiss, R. F. & Price, B. A. Nitrous oxide solubility in water and seawater. *Mar. Chem.* 8, 347–359 (1980).

60. Dlugokencky, E. J., Masarie, K. A., Lang, P. M. & Tans, P. P. Continuing decline in the growth rate of the atmospheric methane burden. *Nature* 393, 447–450 (1998).

61. Dunne, J. P., Armstrong, Ra, Gnnadesikan, A. & Sarmiento, J. L. Empirical and mechanistic models for the particle export ratio. *Glob. Biogeochem. Cycles* 19, 1–16 (2005).

62. Kretschmer, K., Biastoch, A., Rüpke, L. & Burwicz, E. Modeling the fate of methane hydrates under global warming. 610–625, https://doi.org/10.1002/2014GB005011 (2015).

63. Sarmiento, J. L. *Ocean Biogeochemical Dynamics* (Princeton University Press, 2013).

64. Wanninkhof, R. Relationship between wind-speed and gas-exchange over the ocean. *J. Geophys. Res.-Oceans* 97, 7373–7382 (1992).

65. Nightingale, P. D. et al. In situ evaluation of air-sea gas exchange parameterizations using novel conservative and volatile tracers. *Glob. Biogeochem. Cycles* 14, 373–387 (2000).

66. Liss, P. S. & Merlivat, L. in *The Role of Air–Sea Exchange in Geochemical Cycling* (ed. Patrick Buat-Ménard) 113–127 (Springer, Netherlands, 1986).

67. Atlas, R. et al. A cross-calibrated, multiplatform ocean surface wind velocity product for meteorological and oceanographic applications. *Bull. Am. Meteorol. Soc.* 92, 157–174 (2011).

68. Ostrovsky, I., Mcginnis, D. F., Lapidus, L. & Eckert, Quantifying gas ebullition with echosounder: the role of methane transport by bubbles in a medium-sized lake. *Limnology and Oceanography: Methods* 6, 105–118 (2008).

69. Lana, A. et al. An updated climatology of surface dimethylsulfide concentrations and emission fluxes in the global ocean. *Glob. Biogeochem. Cycles* 25, https://doi.org/10.1029/2010gb003850 (2011).

70. Weber, T., Cram, J. A., Leung, S. W., DeVries, T. & Deutsch, C. Deep ocean nutrients imply large latitudinal variation in particle transfer efficiency. *Proc. Natl Acad Sci USA* 113, 201604414 (2016).

## Acknowledgements
T.W. and N.A.W. were supported by NASA-IDS grant NNX17AK11G. A.K. was supported by the Baltic Earth joint project BONUS INTEGRAL (FKZ03F0773B). We thank all data collectors that contributed to the MEMENTO archive, and D. McGinnis for providing bubble model output. We also acknowledge the Ocean Carbon and Biogeochemistry Project Office for facilitating a workshop at Lake Arrowhead in October 2018, which facilitated interactions among the ocean methane community and greatly improved this work.

## Author contributions
T.W. designed the study. A.K and N.A.W. assembled the methane supersaturation dataset. N.A.W. and T.W. developed the statistical mapping method and generated methane emissions estimates. T.W. wrote the paper with input from N.A.W. and A.K.

## Competing interests
The authors declare no competing interests.
