## [Peer Review File · Nature Communications]

Reviewers' comments:

Reviewer #1 (Remarks to the Author):

General Comments

This is an interesting, timely and possibly very important paper on oceanic emissions of methane from the surface ocean. The authors use a recently curated data set on global oceanic methane concentrations and state-of-the-art machine-learning statistical models to map the disequilibrium of methane (the signed difference from equilibrium with the atmosphere) to derive a revised global flux of methane from the ocean to the atmosphere. Their revised estimate, 6-12 Tg/year, is at the lower end of contemporary estimates but has a much lower range (and uncertainty). More importantly, they provide new insights as to the pathways leading to methane supersaturations in the surface ocean, suggesting that net primary production and aerobic methanogenesis from recycling of newly formed organic matter may be a major source of methane. Given the important role of methane as a potent greenhouse gas, and the fact that aerobic methanogenesis – discovered only a decade ago – was not even considered as a source of methane in recent budgets, this paper presents a bold hypothesis for future investigation.

Specific Comments

1. lines 78-80: The authors state that the models/methods used “recognize pattern similarities...” but how can they “fill gaps” where no data currently exist? The surface ocean is grossly undersampled with respect to methane and other biogenic gases. Furthermore, a recent SCOR working group has conducted a laboratory intercomparison and found, in some cases, very poor agreement among “leading laboratories.” The Wilson et al. (2018, *Biogeosciences* 15: 5891-5907) paper should be cited.
2. line 95: used to propagate
3. line 98: What is meant by “broadly resembles?”
4. line 103; Why is the Southern Ocean so different?
5. line 202: “...specifically phosphonate compounds.” Only methylphosphonate (one of many possible “phosphonate” compounds) could lead to the production of methane.
6. The hypothesis that NPP leads to organic matter cycling and methane production is not entirely consistent with an aerobic methanogenesis pathway. If methylphosphonate oxidation is the source of methane, then the greatest amount of methane should be produced where gross primary production is high, but net primary production is low, zero or negative, indicating that the greatest amount of organic matter has been oxidized. Net primary production can vary independently of gross primary production, and it is not clear to this reviewer whether the currently employed satellite-based models really estimate NPP, or some other property. Finally, the use of WOA13 to estimate surface phosphate should be used with extreme caution since that data base does not distinguish between samples analyzed with “standard” colorimetric assay vs. high sensitivity methods that can resolve the phosphate concentrations from the oligotrophic waters that dominate tropical and subtropical habitats worldwide.

Reviewer #2 (Remarks to the Author):

General comments

The authors report a tremendous effort using advanced interpolation tools to produce an objective mapping of CH₄ in the ocean based on a global data-base of CH₄. This is an exciting and timely effort and generally confirms previous global estimates based on much more primitive upscaling that the open ocean is an extremely marginal player in the global CH₄ budgets, while coastal waters emit distinctly more than the open ocean.

Specific comments

L 19: Here and elsewhere in the ms specify Tg CH₄ instead of Tg

L45: "stripping" is an awkward term I suggest to replace by dissolution

L47: I'm not sure "recently" applies here since both the cited references are 11 yrs old.

L133: The cited emission from the Arctic includes ebullitive CH₄ emissions, which could explain the difference.

L200-203: This is not correct. Karl et al. (2008) showed that CH₄ is produced when methylphosphonate is added in large quantities to seawater. This molecule is artificially produced for industrial applications, and there is very little evidence that it occurs naturally in oceans (or elsewhere on Earth). There's no evidence at all that it is produced by phytoplankton. Also, according to Karl et al. (2008), the degradation of methylphosphonate by micro-organisms is supposed to occur in P depleted waters where phytoplankton production should be extremely low. So the work of Karl et al. (2008) does not allow to explain the relation between CH₄ and NPP. In fact, the relation between CH₄ and NPP tends to disprove the hypothesis of Karl et al. (2008), something that is quite interesting and should be mentioned in text.

L208-211: Other hypothesis can explain the relation between CH₄ and NPP. Higher NPP could lead to more aggregates or zooplankton fecal pellets leading to more CH₄ according to Karl and Tilbrook (1994). CH₄ could also be produced by transformations of DMS(P,O) (Florez-Leiva et al. 2013). To test this, I suggest the authors test a correlation between CH₄ and the DMS climatology of Lana et al. (2011). Finally, according to the lab experiments of Lenhart et al. (2016), phytoplankton itself can produce CH₄.

So there are several ways to interpret the relation between CH₄ and NPP. In fact, the relation could be indirect and reflect for instance different mixing regime. Highly stratified waters have a low NPP and possible higher methane oxidation due to higher temperatures. While more mixed conditions will stimulate NPP and possibly bring higher CH₄ concentrations from depth or correspond to lower temperatures, leading to the low methane oxidation that seems to be very low at temperatures < 10°C (Dunfield et al. 1993).

Did you test if the relation between CH₄ and NPP also occurs in coastal waters ? At least locally there seems to be a relation between CH₄ and chlorophyll-a in some coastal sites (Borges et al. 2018). Overall, the discussion of the drivers of CH₄ in coastal water is inexistent, and all of the discussion focusses on open ocean that has a much lower emission rate. Some readers will be interested to know what "nice" correlations can emerge with CH₄ in coastal waters (with depth ? NPP ?).

It could be useful to provide in a table with the deltaCH₄ and flux per ocean basin, some readers might find this useful.

You should consider adding to the error analysis the uncertainty of dissolved CH₄ concentration. MEMENTO aggregates data from numerous groups that leads to substantial uncertainty has shown by the recent intercalibration reported by Wilson et al. (2018).

L227-229: I'm not sure the relation between CH₄ and NPP can be used to predict the future evolution of oceanic CH₄ emissions. The future decrease of NPP is supposed to be related to a decrease of nutrient inputs due to an increase of stratification. This increase of stratification should lead to a decrease of O₂ in the ocean interior and the extension of hypoxia/anoxia zones that might stimulate the production of CH₄. Please note that long-term time-series do not show a systematic decrease of NPP in the ocean (Chavez et al. 2011), so the postulated future decrease of oceanic PP is open to debate.

L367: Incomplete reference

L636: It's very regretful that the monthly climatologies are not publically available. This will probably reduce the impact of paper. Also, the MEMENTO data base was a community effort, it is regretful and somewhat unfair that this community cannot freely access the resulting climatologies.

Refs

Borges, A.V., Speeckaert, G., Champenois, W., Scranton, M.I., Gypens, N., 2018. Productivity and temperature as drivers of seasonal and spatial variations of dissolved methane in the Southern Bight of the North Sea. *Ecosystems* 21, 583-599.

Chavez FP, Messié M, Pennington JT. Marine primary production in relation to climate variability and change. *Ann Rev Mar Sci.* 2011;3:227-60.

Dunfield, P., R. Knowles, R. Dumont, and T. R. Moore (1993), Methane production and consumption in temperate and subarctic peat soils: Response to temperature and pH, *Soil Biol. Biochem.*, 25, 321-326.

Florez-Leiva L, Damm E, Farías L. 2013. Methane production induced by dimethylsulfide in surface water of an upwelling ecosystem, *Progress in Oceanography*, 112-113, 38-48.

Karl DM, Tilbrook BD 1994. Production and transport of methane in oceanic particulate organic matter. *Nature*, 368, 732-734.

Lana, A., et al. (2011) An updated climatology of surface dimethylsulfide concentrations and emission fluxes in the global ocean, *Global Biogeochem. Cycles*, 25, GB1004, doi:10.1029/2010GB003850.

Lenhart K, Klintzsch T, Langer G, Nehrke G, Bunge M, Schnell S, Keppler F. 2016. Evidence for methane production by marine algae (*Emiliana huxleyi*) and its implication for the methane paradox in oxic waters. *Biogeosciences*, 13, 3163-3174

Wilson ST, HW Bange, DL Arévalo-Martínez, J Barnes, AV Borges, I Brown, JL Bullister, M Burgos, DW Capelle, M Casso, M de la Paz, L Farías, L Fenwick, S Ferrón, G Garcia, M Glockzin, DM. Karl, A Kock, S Laperriere, CS. Law, CC Manning, A Marriner, J-P Myllykangas, JW. Pohlman, A P. Rees, AE. Santoro, M Torres, PD. Tortell, RC Upstill-Goddard, DP. Wisegarver, GL Zhang, G Rehder (2018) An intercomparison of oceanic methane and nitrous oxide measurements, *Biogeosciences*, 15, 5891-5907.

Alberto V. Borges
University of Liège

Reviewer #3 (Remarks to the Author):

Summary response

The authors present a global emissions estimate of oceanic methane using two established machine learning approaches and a new approach to ebullition (bubbling) fluxes. They first build a new compilation of observational data, using the existing community's published compilation too. The manuscript then focuses on uncertainty, leading an interpretation of the relative sources of error from different regions and input variables. This leads to a better constraint on global emissions for global modelling and gives also direction to future work in the observational community. The whole approach taken provides a "best guess" based on the available data, which here has been coupled with a persuasive and detailed exploration of the uncertainties. The manuscript is well prepared and scientifically novel and I would recommend its publication following responses to minor comments made below.

Specific comments

I am surprised by the use of a derived parameter as a feature for prediction (NPP) when the input variables used to make this are surely available as gridded products (e.g. phytoplankton carbon and chlorophyll). Considering the algorithms used wouldn't these same results be expected if the core drivers behind a derived variable were provided?

There is some discussion of resolution effects which is hard to follow. If input data of higher resolution was available (e.g. gridded variables at a scale of $\sim 10 \times 10$ km or smaller - e.g. chlorophyll is available at least $\sim 4 \times 4$ km), would this not be expected to better capture the heterogeneity seen at these scales in coastal resolution? The argument for improvements plateauing at a horizontal resolution of 0.25×0.25 degree ($\sim 25 \times 25$ km) seems too much based on the input data, for which the same resolution was chosen by the authors.

The reference Schmidtko et al 2013 (#45) says the oceanographic variables are only at 0.5×0.5 deg horizontal resolution? There are finer products out there, such as the World Ocean Atlas (at least 0.25×0.25). So if the products used were indeed 0.5×0.5 deg, why were not finer products used?

The data preparation seems to have been focused around one of the algorithms used, artificial neural networks (ANN). However, the analysis shows the resulting performance of the two algorithms to be comparable. Questions are therefore raised from this about what difference in performance would be seen from the Random Forest (RF) approach taken if different approaches were taken (see specific comments below on transforming data/mapping point data). Regardless, this is a fairly minor nuance and would likely offer only small improvements in the results. Some small mention of the points raised would be sufficient and I would not argue that this should affect my recommendation to published.

Line 62 - Why have the data been mapped to a coarse resolution instead of just using the sparse data? Use of point data would be more appropriate for the Random Forest technique used here and give more data to work with. Was this due to optimising the impacts to use with the ANN technique?

Line 77 - I assume one of the references here if meant to be for a paper with some of the same authors on sparse marine data (not their one on chemistry integration cited)?

e.g.

This one:

Keller, C. A. & Evans, M. J. Application of random forest regression to the calculation of gas-phase chemistry within the GEOS-Chem chemistry model v10. *Geosci. Model Dev.* 12, 1209-1225, doi:10.5194/gmd-12-1209-2019 (2019).

Should be:

Sherwen, T., Chance, R. J., Tinel, L., Ellis, D., Evans, M. J., and Carpenter, L. J.: A machine learning based global sea-surface iodide distribution, *Earth Syst. Sci. Data Discuss.*, <https://doi.org/10.5194/essd-2019-40>, in review, 2019.

Line 126 - Please see the earlier comment. Although the methods say the effect of the mapping of point data to a coarser map was considered, I am not sure whether the effect of the input variables (features) being gridded to 0.25x0.25 was? If high-resolution variable data were considered (e.g. chlorophyll etc are available to 4km) then this statement would be more persuasive.

Line 185 - subscript missing for delta methane

Line 194 - Please see the earlier comment. Why not just use the inputs for the algorithm (phytoplankton carbon and chlorophyll) instead of providing a product derived from them? Should this just lead to including errors from the other approach? Shouldn't the algorithms used here be able to pick this the parts of the inputs for the NPP prediction that are important for delta methane?

Line 203-207 - The exploration of contributions here is interesting. However, as the RF used is interoperable could not more information about the drivers be extracted directly? The use of multiple linear regression here does not seem like the most comprehensible choice for the reader.

Line 375 - Please state the number of observations that were binned into the coarser mapped product.

Line 385 - Why were only 100 ANN and RRF shown in fig 2c? Aesthetic reasons? Were they just choose as random sample? Please state in the caption.

Line 440 - Was the temperature etc with each observation also mapped to a coarser (0.25x0.25) resolution?

Line 505 - Why was a transform used for the data used for the RF? Surely this is not needed. What impact on prediction is seen if this is not done?

Line 513 - Please state whether any further randomisation was used (e.g. bootstrap aggregation)?

Line 569 - Please give approximate grid resolution in brackets for readability.

Line 638 - Please consider archiving your output data in a permanent data repository and providing a DOI for it here.

General Comment to all reviewers

We are pleased that all three reviewers are supportive of our work, and find that it makes a timely and important contribution to the field. At the same time, each reviewer provided constructive criticism aimed at improving the paper, for which we are grateful. We have taken their comments seriously and have addressed them to the best of our ability in the revised manuscript and point-by-point responses below. Where the reviewers have suggested potential modifications to our machine learning and error propagation methods, we have taken the approach of conducting sensitivity tests based on small model ensembles (generally 200 members) trained using the modified methods. Recreating the entire ensemble of 200,000 models used in our main results would delay our work by many months, without fundamentally changing our results, as demonstrated in the sensitivity tests. These new sensitivity tests have resulted in new supplementary figures, including two new panels in Fig. S1 and the new Fig. S4. They have also further demonstrated the robustness of our results, and we thank the reviewers for suggesting them.

We are also pleased that all three reviewers found the discussion of open ocean methane production mechanisms interesting, and agree that there are many additional steps that could be taken to expand on this. However, we wish to keep the focus of the current manuscript squarely on constraining oceanic methane emissions, which is already a lot of ground to cover in a single paper given the different considerations required for diffusive and ebullitive emissions. Our interpretation of the ΔCH_4 distribution is therefore necessarily brief and focused on the first order open-ocean pattern (i.e. the relationship to NPP and its potential causes). A more detailed examination of controls on ΔCH_4 is the focus of the next stage of our research, utilizing a combination of statistical and mechanistic models, and will be discussed in a dedicated follow-up manuscript that can better do it justice. Nevertheless, we have taken the reviewers' points seriously, and have made sure to add the alternative hypotheses and appropriate caveats they raised to our interpretation section.

Finally, we wish to make the reviewers aware of a mistake we found in the text during our revision. The "likely" ranges that we cite for integrated fluxes (e.g. 2-6Tg/yr for the global diffusive flux) are actually the 10th-90th percentile ranges of the probability distributions, but were incorrectly referred to as 90% confidence intervals, which is a holdover from an earlier version of our work where that statistic was used. Because the probability distributions derived by our Monte Carlo procedure are positively skewed with very long tails, the 5th-95th percentile range can be substantially wider at the upper end (2-8Tg/yr for global flux), but this poorly reflects where the "weight" of the probability distribution lies. We therefore decided to use the 10th-90th percentiles, which trim the long tail and provide a more meaningful range. We have updated all instances in the text and apologize to the reviewers for the discrepancy in the previous version.

Response to Reviewer #1 (Reviewer comments in blue)

This is an interesting, timely and possibly very important paper on oceanic emissions of methane from the surface ocean. The authors use a recently curated data set on global oceanic methane concentrations and state-of-the-art machine-learning statistical models to map the disequilibrium of methane (the signed difference from equilibrium with the atmosphere) to derive a revised global flux of methane from the ocean to the atmosphere. Their revised estimate, 6-12 Tg/year, is at the lower end of contemporary estimates but has a much lower range (and uncertainty). More importantly, they provide new insights as to the pathways leading to methane supersaturations in the surface ocean, suggesting

that net primary production and aerobic methanogenesis from recycling of newly formed organic matter may be a major source of methane. Given the important role of methane as a potent greenhouse gas, and the fact that aerobic methanogenesis – discovered only a decade ago – was not even considered as a source of methane in recent budgets, this paper presents a bold hypothesis for future investigation.

We thank the reviewer for recognizing the timeliness and importance of our work. We agree that a more accurate, up-to-date estimate of marine methane emissions is sorely needed for inclusion in atmospheric budgets, and we are pleased that the reviewer finds our new approach takes a major step towards that goal.

Lines 78-80: The authors state that the models/methods used “recognize pattern similarities...” but how can they “fill gaps” where no data currently exist? The surface ocean is grossly undersampled with respect to methane and other biogenic gases.

We apologize that this description of the machine learning methods was unclear to the reviewer. Essentially, these algorithms build a highly nonlinear statistical model for ΔCH_4 as a function of predictor variables. The model can then be used to extrapolate ΔCH_4 into regions where no data exists, as long as data does exist for all predictor variables. Just as a much simpler linear correlation between ΔCH_4 and NPP could be used to extrapolate ΔCH_4 into any region with NPP data. While we need to keep this description brief in the text, we have opted to replace the sentence with the following one, which we believe is more straightforward:

Line 82: These methods build nonlinear statistical models for ΔCH_4 based on its relationship to physical and biogeochemical “predictor” variables, whose distributions are well known and are plausibly linked to ΔCH_4 (see Methods), allowing global extrapolation of ΔCH_4 in the mixed layer (Fig. 2a,b).

Furthermore, a recent SCOR working group has conducted a laboratory intercomparison and found, in some cases, very poor agreement among “leading laboratories.” The Wilson et al. (2018, *Biogeosciences* 15: 5891-5907) paper should be cited.

We thank this reviewer and Reviewer #2 for bringing this to our attention. The interlaboratory spread has now been acknowledged and the Wilson et al. paper cited in two different contexts. First, Reviewer #2 was curious about how this potential source of error might bias our global flux calculations. While this is difficult to assess, we devised a sensitivity test that is fully described in the response to Reviewer #2, and we invite this reviewer to read it. We found that our global flux estimates are relatively insensitive as long as the database as a whole is not systematically biased, and this is demonstrated in a new supplementary figure (now Fig. S4). Second, in our final “Outlook” section, where we make suggestions for the community moving forwards, we have highlighted the importance of further examining and resolving the poor agreement between laboratories, by adding the following sentence:

Line 227: Understanding and resolving interlaboratory discrepancies in $[\text{CH}_4]$ measurements²⁸ should also be prioritized, so that consistent data may be synthesized across multiple sources.

Line 95: used to propagate

Thanks for catching this – text has been updated.

Line 98: What is meant by “broadly resembles?”

We mean that the spatial pattern of the flux is qualitatively similar to the pattern of ΔCH_4 , i.e. suggesting that ΔCH_4 rather than wind speed controls the pattern of the flux. We agree with the reviewer that the current wording is imprecise, and have updated to:

Line 104: The spatial pattern of air-sea flux predicted by these model ensembles is qualitatively similar to the ΔCH_4 distribution, with highest fluxes in shallow shelf regions that often exceed rates of $10\text{mmol/m}^2/\text{yr}$ (Fig. 3, Table S1).

Line 103; Why is the Southern Ocean so different?

We have restricted this portion of the text to description of the ΔCH_4 distribution, and we withhold interpretation for the section “Open ocean methane production”. As we state there, we interpret the undersaturation in the Southern Ocean as reflecting the upwelling of old, deep water that has undergone methane oxidation. One would expect upwelling water to exhibit this signal, unless there is strong *in situ* source in the surface ocean.

Line 202: “...specifically phosphonate compounds.” Only methylphosphonate (one of many possible “phosphonate” compounds) could lead to the production of methane.

We have clarified this by revising the following sentence, which has also been expanded to include the alternative methanogenesis pathway raised by Reviewer #2:

Line 207: This suggests a linkage between organic matter cycling and CH_4 production, supporting proposed aerobic pathways of methanogenesis that include microbial degradation of methylphosphonate in dissolved organic matter^{12,44}, and direct production during coccolithophore growth¹⁴.

The hypothesis that NPP leads to organic matter cycling and methane production is not entirely consistent with an aerobic methanogenesis pathway. If methylphosphonate oxidation is the source of methane, then the greatest amount of methane should be produced where gross primary production is high, but net primary production is low, zero or negative, indicating that the greatest amount of organic matter has been oxidized. Net primary production can vary independently of gross primary production, and it is not clear to this reviewer whether the currently employed satellite-based models really estimate NPP, or some other property.

We appreciate the reviewer’s concern, but believe they are using a definition of NPP that does not align with our own. The satellite algorithms we use estimate NPP defined as gross primary production minus autotroph respiration. It is therefore a measure of the supply of organic matter that for oxidation by heterotrophic bacteria and higher trophic levels. The correlation of ΔCH_4 against NPP is therefore a reasonable indicator of potential methanogenesis during organic matter oxidation. The definition used by the reviewer aligns more closely with “net community production” (NCP), which is NPP minus heterotroph respiration. We agree that NCP can be decoupled from NPP, and would be a poor indicator of organic matter oxidation in the surface. In fact, over a seasonal averaging window NCP should be closely related to the POC export flux, which we did include as predictor variable in our model, but is much more weakly correlated to NPP (Table S2)

Finally, the use of WOA13 to estimate surface phosphate should be used with extreme caution since that data base does not distinguish between samples analyzed with “standard” colorimetric assay vs. high sensitivity methods that can resolve the phosphate concentrations from the oligotrophic waters that dominate tropical and subtropical habitats

worldwide.

We agree with the reviewer that WOA13 contains phosphate measurements from different methodologies, that may introduce errors to the global distribution. However, given previous hypotheses concerning methane production under P limitation, we felt that phosphate was important to include as a predictor variable, and WOA13 represents the best available global database at this time. Each predictor variable we use is likely subject to its own uncertainties and methodological biases, and we agree with the reviewer that it is important to be explicit about these limitations. We have therefore added the following caveat in the Methods section:

Line 502: We note that while we have chosen the most up-to-date global data products for use in our work, each is likely subject to its own uncertainties or methodological biases, and some have been subjected

Response to Reviewer #2 (Reviewer comments in blue)

The authors report a tremendous effort using advanced interpolation tools to produce an objective mapping of CH₄ in the ocean based on a global data-base of CH₄. This is an exciting and timely effort and generally confirms previous global estimates based on much more primitive upscaling that the open ocean is an extremely marginal player in the global CH₄ budgets, while coastal waters emit distinctly more than the open ocean.

We thank the reviewer for recognizing the magnitude of our effort – indeed this manuscript is the culmination of years of work! We share their excitement about replacing previous estimates from “primitive upscaling” in global budgets, and refocusing the communities attention on coastal waters that dominate the air-sea flux.

L 19: Here and elsewhere in the ms specify Tg CH₄ instead of Tg

We thank the reviewer for pointing out that this was unclear. We have changed all instances of Tg/yr in the abstract to TgCH₄/yr. However, because this unit is used many times throughout the paper and in figure axes, for brevity and readability we have chosen to maintain Tg/yr but to define at first usage exactly what we mean when we use this unit:

Line 40: The global ocean is a highly uncertain term in the atmospheric CH₄ budget, emitting 5-25Tg of CH₄ per year (hereafter Tg/yr) or 1-13% of all natural emissions⁴.

L45: “stripping” is an awkward term I suggest to replace by dissolution

Thanks, we have made this change.

L47: I’m not sure “recently” applies here since both the cited references are 11 yrs old.

We agree, and have changed to “more recently” in order to draw a contrast to the traditional view that the seafloor was the ultimate source of all marine methane. We have also updated this sentence to include another reference brought to our attention to the reviewer, and to more clearly reflect the fact that while these pathways have identified, they have not been proven to produce methane in the marine environment:

Line 50: More recently, aerobic methanogenesis pathways have been identified that may produce CH₄ *in situ* in the surface ocean mixed layer, providing a more direct conduit to atmosphere¹²⁻¹⁴

L133: The cited emission from the Arctic includes ebullitive CH₄ emissions, which could explain the difference.

While this reference does consider both diffusive and ebullitive fluxes, we made sure to conduct an apples-to-apples comparison, by only comparing to the diffusive flux estimate. The number we cite (3.3Tg/yr) is the sum of summertime (0.9Tg/yr) and wintertime (2.4Tg/yr) diffusive fluxes reported in the reference.

L200-203: This is not correct. Karl et al. (2008) showed that CH₄ is produced when methylphosphonate is added in large quantities to seawater. This molecule is artificially produced for industrial applications, and there is very little evidence that it occurs naturally in oceans (or elsewhere on Earth). There's no evidence at all that it is produced by phytoplankton. Also, according to Karl et al. (2008), the degradation of methylphosphonate by micro-organisms is supposed to occur in P depleted waters where phytoplankton production should be extremely low. So the work of Karl et al. (2008) does not allow to explain the relation between CH₄ and NPP. In fact, the relation between CH₄ and NPP tends to disprove the hypothesis of Karl et al. (2008), something that is quite interesting and should be mentioned in text.

We agree with the reviewer that Karl et al. (2008) added artificial methylphosphonate (MPn) to their incubations. However, the newer work by Repeta et al. (2016) that we cite identifies this compound as an important constituent of naturally-occurring marine DOM, and demonstrates methanogenesis in incubations amended with natural DOM rather than artificial MPn. They conclude that cycling of the natural MPn inventory could explain observed surface ocean methane supersaturation. We also note that Karl et al. (2008) show that methanogenesis proceeds (although at a slower rate) at ambient phosphate levels up to 1μM, meaning this process needn't be restricted to oligotrophic regions, and Repeta et al (2016) hypothesize that it may be an important ubiquitous source methane. Overall then, it appears that methanogenesis during MPn cycling should be positively related to the supply of organic matter (NPP) and negatively related to phosphate availability, both of which are borne out by our correlation analysis (Table S2). We therefore feel justified to state that the distribution of ΔCH₄ is broadly consistent with the MPn hypothesis, although in response to this reviewer's other comments (see below) we have been more careful to mention the alternative hypotheses and caveats.

L208-211: Other hypothesis can explain the relation between CH₄ and NPP. Higher NPP could lead to more aggregates or zooplankton fecal pellets leading to more CH₄ according to Karl and Tilbrook (1994). CH₄ could also be produced by transformations of DMS(P,O) (Florez-Leiva et al. 2013). To test this, I suggest the authors test a correlation between CH₄ and the DMS climatology of Lana et al. (2011). Finally, according to the lab experiments of Lenhart et al. (2016), phytoplankton itself can produce CH₄. So there are several ways to interpret the relation between CH₄ and NPP.

We are grateful to the reviewer for pointing us towards the work of Lenhart et al. (2016), which we were not aware of. We find their results very compelling, and agree that this is an equally viable explanation of the relationship between NPP and ΔCH₄. We have therefore amended the following sentence in the text to place this on even footing with the MPn hypothesis:

Line 207: This suggests a linkage between organic matter cycling and CH₄ production, supporting proposed aerobic pathways of methanogenesis that include microbial degradation of methylphosphonate in dissolved organic matter^{12,44}, and direct production during coccolithophore growth¹⁴.

We have also removed the discussion of a secondary relationship to phosphate from this section of the text. While this did emerge from the Multiple Linear Regression analysis, it was marginal and does not make a large improvement over the correlation to NPP alone. We are concerned that by discussing this secondary relationship, it will appear that we are favoring the MPn hypothesis over the coccolithophore hypothesis, and we would prefer to remain agnostic and impartial on this front.

The reviewer also mentions two alternative methanogenesis pathways: anaerobic production in sinking organic particles (Karl & Tilbrook 1994), and production during DMS transformations (e.g. Florez-Leiva et al., 2013). The first of these mechanisms would be expected to result in a stronger correlation of ΔCH_4 to the sinking POC flux than to NPP, which we do not find in our analysis (Table S2). We were aware of the DMS climatology of Lana et al. (2011), but did not originally include this as a predictor variable in our machine learning algorithms. This is because this climatology is itself based on a relatively sparse database that has been gap-filled by objective analysis (approximately an order of magnitude less datapoints than World Ocean Atlas). This introduces the potential for mapping errors in the predictor variable to propagate into our ΔCH_4 maps. Nevertheless, we have obtained the DMS climatology and conducted the correlation analysis as suggested by the reviewer. We find $R^2=0.04$ when DMS is correlated against the mapped ΔCH_4 product, and $R^2=0.008$ against raw ΔCH_4 data, which are much lower than for NPP (0.3 and 0.12 respectively).

We have not discussed the correlations against POC flux or DMS in the main text, because our interpretation of the ΔCH_4 distribution is necessarily very brief. As discussed in the General Comment above, the focus of this paper is squarely on providing a more robust estimate of marine methane emissions. We included the brief interpretation of open-ocean ΔCH_4 because we know it will be of interest to the community in light of recently discovered methanogenesis pathways, but have restricted the discussion to the first order pattern (i.e. the correlation to NPP). The weaker correlations against other variables (like POC flux and DMS) are all interesting in their own right, but a full discussion of each is beyond the scope of this paper. The next stage of our project will attempt to constrain methane production rates and mechanisms using a combination of statistical and mechanistic models, and we will be able to provide a more complete evaluation of each hypothesis at that time.

In fact, the relation could be indirect and reflect for instance different mixing regime. Highly stratified waters have a low NPP and possible higher methane oxidation due to higher temperatures. While more mixed conditions will stimulate NPP and possibly bring higher CH₄ concentrations from depth or correspond to lower temperatures, leading to the low methane oxidation that seems to be very low at temperatures < 10°C (Dunfield et al. 1993).

If mixing and/or oxidation were the dominant causes of the ΔCH_4 vs. NPP relationship, then one might expect a stronger correlation to salinity or temperature than to NPP. However, the point raised by the reviewer is a very important one: because our model is statistical and not mechanistic, and does not include any information about the physical CH₄ supply, we cannot conclude that the ΔCH_4 vs. NPP correlation is causative and not coincidental. We share the reviewer's concern that the previous text came across too definitively as supporting the aerobic methanogenesis hypothesis, and have therefore added this important caveat at the end of the interpretation section:

Line 213: However, we cannot definitively conclude that this relationship arises mechanistically from methanogenesis, and not from spatial variations in CH₄ oxidation or the physical CH₄ supply, which may also be correlated with NPP.

Did you test if the relation between CH₄ and NPP also occurs in coastal waters ? At least locally there seems to be a relation between CH₄ and chlorophyll-a in some coastal sites (Borges et al. 2018). Overall, the discussion of the drivers of CH₄ in coastal water is inexistent, and all of the discussion focusses on open ocean that has a much lower emission rate. Some readers will be interested to know what “nice” correlations can emerge with CH₄ in coastal waters (with depth ? NPP ?).

We have limited our interpretation of the ΔCH_4 distribution to the open ocean, because this is the only region that it is well constrained enough to conduct a meaningful analysis. In the open ocean, each member of our model ensemble generates a similar ΔCH_4 distribution, which is reflected in the narrow uncertainty range of diffusive CH₄ emissions in this region (Fig. 4a). In the open ocean, R² for the correlation against NPP is 0.29 for ensemble-mean ΔCH_4 , and ranges between 0.23-0.3 for individual ensemble members, showing that the correlation is strong and robust. On the other hand, the coastal ΔCH_4 distribution is not well constrained (different ensemble members generate quite different distributions), which is reflected in the wide uncertainty range in our coastal emissions estimates (Fig. 4a). In these region, R² for the correlation against NPP is 0.04 for the ensemble-mean, and 0.0005-0.09 for individual ensemble members. In other words, the correlation is weak and not robust, because the range is much wider in a relative sense. While we agree that correlations in some locations (e.g. Borges et al., 2018) are exciting and compelling, we cannot use our model to argue for the global applicability of these relationships. Instead, we aim to deliver a simple message to the community regarding coastal waters: our emissions estimates will be improved by collecting new datasets that accurately resolve sharp coastal gradients, to better constrain the ΔCH_4 distribution. We see the Borges et al. (2018) study as a prime example of this, which is why it was cited here:

Line 221: The majority of the remaining uncertainty in our estimate is attributed to shallow near-shore environments, where ΔCH_4 and diffusive emissions vary most among our model ensembles (Fig. 4a), and where relatively unconstrained ebullitive fluxes are concentrated (Fig. 5a). To further refine our estimate, future observational efforts should focus on these shallow environments and sample with the resolution to capture sharp coastal gradients in ΔCH_4 ²², while employing new imaging technologies³⁷ to further constrain bubble dynamics and ebullition.

It could be useful to provide in a table with the deltaCH₄ and flux per ocean basin, some readers might find this useful.

We agree with the reviewer that some readers will be interested in how mean ΔCH_4 and integrated air-sea flux vary between ocean basins. We have now provided such a table as Table S1, which summarizes these properties from our ensemble-mean climatology (previous Table S1 has moved to S2). There were two additional benefits of adding this new table. First, we can refer to it illustrate the mismatch between ΔCH_4 and flux in the Arctic (due to ice cover) that we mention in the text. Second, we can refer to it when we state the total flux from the Arctic Ocean, instead of adding dots to Fig. 4b as we did in the original manuscript. We felt this overcomplicated the figure, which is supposed to illustrate global emissions only, so referring to the table is a cleaner way to make this point.

You should consider adding to the error analysis the uncertainty of dissolved CH₄ concentration. MEMENTO aggregates data from numerous groups that leads to substantial uncertainty has shown by

the recent intercalibration reported by Wilson et al. (2018).

We thank this reviewer and Reviewer #1 for bringing this to our attention. We agree that the interlaboratory spread in CH₄ is concerning and adds another potential source of uncertainty to our analysis. However, it is difficult to account for in a rigorous way because the MEMENTO database does not contain an uncertainty estimate for each individual observation, and even if it did, these would likely not adequately reflect the biases between laboratories. Nevertheless, we have attempted to devise a sensitivity test to assess the impact of these observational uncertainties on our flux estimates. Wilson et al. (2018) show that measurements from individual laboratories can diverge from the interlaboratory mean by up to 25% in strongly supersaturated waters and up to 50% in weakly supersaturated waters. (The divergence can be larger when in regions where CH₄ is close to saturation, but such regions do not drive significant flux).

We therefore conducted a new ensemble of simulations in which “true” [CH₄] is assumed to lie within the interval $(1-R.E.) \cdot [CH_4]_{obs}$ to $(1+R.E.) \cdot [CH_4]_{obs}$, where $[CH_4]_{obs}$ is reported measured value, and R.E. is the relative error. A selection within this range is made for each individual observation before the global climatology is formed and mapping method applied. This new sensitivity test is now described in the methods section, and the results shown in the new supplementary Fig. S4, which is reproduced below for the reviewer’s convenience. We find that when R.E.=0.25, our global flux estimate remains largely unchanged, and when R.E.=0.5 the most likely value remains the same (~4Tg/yr) but the 10th-90th percentile range expands to 1.5-6.9Tg/yr (from 2.1-6.3). Therefore, even in the extreme case of up to 50% error in each observation, the impact on the global flux estimates is relatively modest, as long as the database as a whole is not systematically biased.

We have now pointed to this sensitivity test result and others in the main text:

Line 126: Sensitivity tests revealed that the global flux is relatively insensitive to improving the model grid resolution (Fig S1c-d), the choice of biological predictor variables (Fig S1e-f), or propagation of potential measurement errors (Fig. S4, ref 28).

We have also decided to highlight the need to address the issue of interlaboratory divergence in our Outlook section, as follows:

Line 227: Understanding and resolving interlaboratory discrepancies in [CH₄] measurements²⁸ should also be prioritized, so that consistent data may be synthesized across multiple sources.

Fig. S4. Sensitivity of global diffusive methane emissions to relative error in individual [CH₄] measurements.

L227-229: I'm not sure the relation between CH₄ and NPP can be used to predict the future evolution of oceanic CH₄ emissions. The future decrease of NPP is supposed to be related to a decrease of nutrient inputs due to an increase of stratification. This increase of stratification should lead to a decrease of O₂ in the ocean interior and the extension of hypoxia/anoxia zones that might stimulate the production of CH₄. Please note that long-term time-series do not show a systematic decrease of NPP in the ocean (Chavez et al. 2011), so the postulated future decrease of oceanic PP is open to debate.

We are aware that time series sites have not registered systematic declines in NPP, although it is debated whether these records are long enough to discern long-term anthropogenic trends from decadal climate variability. Our statement concerns future climate impacts, and declining NPP does appear to be a robust prediction across climate models (Bopp et al., 2013; Moore et al., 2018). We also agree with the reviewer that changing NPP will have other knock-on effects on ocean biogeochemistry that could impact the marine methane cycle. However, our intention is only to state the most direct implication of the correlation we found between ΔCH_4 and NPP. We have updated this sentence so as not make such a definitive statement, introducing the word "tentatively" and simply stating that future changes in stratification will "impact" marine productivity.

Line 232: The global relationship between ΔCH_4 and NPP reported here provides a simple approach to represent these emissions in coupled ocean-atmosphere models, and tentatively predict future perturbations in this source as ocean warming and stratification impact marine productivity⁴⁵.

L367: Incomplete reference

Corrected, thank you.

L636: It's very regretful that the monthly climatologies are not publically available. This will probably reduce the impact of paper. Also, the MEMENTO data base was a community effort, it is regretful and somewhat unfair that this community cannot freely access the resulting climatologies.

We apologize to the reviewer that our plans for data archiving were not made explicit in the original submission. It is indeed our intention to make the maps of ΔCH_4 and methane flux publicly available in an online repository. However, we were concerned that setting up the archive before submission (so that we could provide a link in the paper) would result in our climatology being used by other researchers before it has passed high bar of peer review. We therefore adopted placeholder language in the original submission. We have now set up an archive on the Figshare repository, and reserved a DOI that will activate as soon as our paper is accepted. The reviewer can access the archive using the following private link: <https://figshare.com/s/a6a338252c4b2a209c3d>.

Response to Reviewer #3 (Reviewer comments in blue)

The authors present a global emissions estimate of oceanic methane using two established machine learning approaches and a new approach to ebullition (bubbling) fluxes. They first build a new compilation of observational data, using the existing community's published compilation too. The manuscript then focuses on uncertainty, leading an interpretation of the relative sources of error from different regions and input variables. This leads to a better constraint on global emissions for global modelling and gives also direction to future work in the observational community. The whole approach

taken provides a “best guess” based on the available data, which here has been coupled with a persuasive and detailed exploration of the uncertainties. The manuscript is well prepared and scientifically novel and I would recommend its publication following responses to minor comments made below.

We thank the reviewer for their supportive comments and for recognizing the novelty of our work. We agree that our approach results in the “best guess” and most rigorous treatment of uncertainty to date for marine methane emissions, which is sorely needed for constructing global atmospheric budgets.

I am surprised by the use of a derived parameter as a feature for prediction (NPP) when the input variables used to make this are surely available as gridded products (e.g. phytoplankton carbon and chlorophyll). Considering the algorithms used wouldn't these same results be expected if the core drivers behind a derived variable were provided?

Our logic for choosing NPP as a predictor is that we strived to select variables that have plausible mechanistic links to ΔCH_4 . We only wanted to select one metric of surface ocean productivity (chlorophyll, plankton biomass, NPP, etc.) and due to previous hypotheses connecting methane production to organic matter oxidation we chose NPP, which is the most direct metric of organic matter supply to heterotrophic community. However, the reviewer raises an excellent point – it is perhaps more logical to select a lower-level variable like chlorophyll (Chl) rather than a derived parameter like NPP, given that the machine learning algorithms should be able to reconstruct any relationship between ΔCH_4 and NPP from the lower level variables that feed into NPP.

We note that to some degree, satellite-derived Chl and organic carbon estimates are themselves derived variables, constructed from various channels of light reflectance and backscatter data. Nevertheless, the reviewer's point is well-taken and we share their curiosity about how replacing NPP with a lower-level biological variable would impact our results. We therefore conducted a sensitivity test in which we generated a new ensemble of ΔCH_4 maps and emissions estimates (100 each from ANN and RRF methods) using the original set of predictor variables but with NPP replaced by Chl. We downloaded the MODIS Chl climatology at 4km spatial resolution from <https://oceancolor.gsfc.nasa.gov/>, and interpolated it to our model grid before model training. We repeated this sensitivity test for a range of model grid resolutions between 0.125° - 2° , and the results have been added to Fig. S1. These new tests therefore serve the dual purpose of addressing this reviewer's concerns about selection of predictor variables, and their other concern about grid resolution, as discussed below.

The results shown in Fig. S1e-f (reproduced below for the reviewer's convenience) demonstrate that using Chl instead of NPP as a predictor variable does not fundamentally change our global flux estimates, or their dependence on model grid resolution.

Fig. S1e-f. Diffusive methane emissions estimated using Chl instead of NPP as a predictor variable, as a function of model grid resolution.

The new sensitivity test is now described in the Methods section, and we point to it (and other tests) in the main text in the following sentence:

Line 126: Sensitivity tests revealed that the global flux is relatively insensitive to improving the model grid resolution (Fig S1c-d), the choice of biological predictor variables (Fig S1e-f), or propagation of potential measurement errors (Fig. S4, ref 28).

There is some discussion of resolution effects which is hard to follow. If input data of higher resolution was available (e.g. gridded variables at a scale of $\sim 10 \times 10$ km or smaller - e.g. chlorophyll is available at least $\sim 4 \times 4$ km), would this not be expected to better capture the heterogeneity seen at these scales in coastal resolution? The argument for improvements plateauing at a horizontal resolution of 0.25×0.25 degree ($\sim 25 \times 25$ km) seems too much based on the input data, for which the same resolution was chosen by the authors.

We would first like to clarify that when we test different grid resolutions, we interpolate predictor variables from their original grids to the new resolution. In other words, the entire process of generating a ΔCH_4 climatology and a set of predictor variables at the same resolution is repeated each time. We apologize if this was unclear and have attempted to clarify. First, we have explicitly noted the original grid resolution of each predictor variable in the “Machine learning mapping” section of the Methods sections. Second, in the sensitivity testing section we have elaborated our procedure:

Line 564: To inform our selection of grid resolution, we applied the full procedure outlined above using grids ranging from 2 to 0.125° in resolution (Fig. S1). In each case, ΔCH_4 data were binned into a climatology at the specified resolution, predictor variables were interpolated to the specified resolution, and an ensemble of 200 ΔCH_4 maps and flux estimates were generated (100 each from ANN and RRF).

However, here and in a later comment, this reviewer has raised a broader and important question that needs to be addressed: does the global flux plateau only plateau at 0.25° resolution simply due to the resolution of the available predictor data? In other words, one would not expect the ΔCH_4 maps generated by machine learning models to change as we move to a 0.125° grid if all the predictors are being interpolated from coarser products and therefore do not differ at 0.125° resolution versus 0.25° resolution. First, it is worth noting that while most predictors we used are only available at 0.25° resolution, bathymetry is available at 0.033° resolution, so interpolation to a 0.125° grid would produce a genuinely higher resolution map than interpolation to 0.25° . Second, while responding to the reviewer’s previous question (use of Chl instead of NPP as a predictor), we took the opportunity to further test the role of predictor resolution. The MODIS Chl climatology we obtained is available at 4km resolution, so again regriding to 0.125° would produce a real improvement in resolution relative to 0.25° . We therefore repeated the grid resolution sensitivity tests using the new set of predictor variables (Chl replaces NPP), and again found that there is no further change in the global flux using a grid finer than 0.25° (in fact the plateau appears to occur at 0.5° in these tests). Given that bathymetry and Chl (which has a distribution similar to NPP) are both important predictors of ΔCH_4 , we find it encouraging that the global flux plateaus at a resolution that is still much coarser than these variables – it suggests that 0.25° is “good enough” to capture the scales of variability in ΔCH_4 . The new sensitivity tests are described in the Methods section and illustrated in Fig 1e-f (reproduced above).

The reference Schmidtko et al 2013 (#45) says the oceanographic variables are only at 0.5×0.5 deg horizontal resolution? There are finer products out there, such as the World Ocean Atlas (at least

0.25x0.25). So if the products used were indeed 0.5x0.5 deg, why were not finer products used?

In our opinion, the MIMOC climatology is the gold standard for mixed layer properties – the curators of that product have done the best possible job of averaging observations over depth through the mixed layer. The World Ocean Atlas climatology is available at higher resolution, but mixed layer averages are not readily available. Nevertheless, we tested whether using the higher resolution WOA13 product would impact our results, by computing our own mixed layer averages of T and S (as we did for phosphate and oxygen) and replacing the MIMOC products in our suite of predictors. In an ensemble of 200 new models trained with these predictors, we find that global flux is indistinguishable from our previous results (~4.1Tg/yr, 2.1-6.3Tg/yr likely range). We elected not to add these sensitivity tests to the manuscript, as they would only serve to lengthen the already expansive Methods section and supplement.

The data preparation seems to have been focused around one of the algorithms used, artificial neural networks (ANN). However, the analysis shows the resulting performance of the two algorithms to be comparable. Questions are therefore raised from this about what difference in performance would be seen from the Random Forest (RF) approach taken if different approaches were taken (see specific comments below on transforming data/mapping point data). Regardless, this is a fairly minor nuance and would likely offer only small improvements in the results. Some small mention of the points raised would be sufficient and I would not argue that this should affect my recommendation to publish.

We agree with the reviewer that data transformation is not necessary for the Random Forest method, and was simply undertaken for consistency between the two methods. We have now explicitly stated this in the “Machine learning mapping” section of the Methods, as requested by the reviewer:

Line 509: While the transformation is not necessary for the RRF method, it was undertaken for operational consistency between our two approaches.

Our reasoning for forming a gridded climatology before model training is outlined in response to the next point.

Line 62 - Why have the data been mapped to a coarse resolution instead of just using the sparse data? Use of point data would be more appropriate for the Random Forest technique used here and give more data to work with. Was this due to optimising the impacts to use with the ANN technique?

We agree that an alternative approach would be to take random draws for model training from the raw sparse dataset, rather than a gridded climatology. The gridding step was added to minimize the effect of heterogeneous data density. Our database contains a few high-resolution cruise tracks, which contribute orders of magnitude more datapoints than the majority of cruises. To illustrate this to the reviewer, we have attached a map of data density (on a 0.25° grid) which illustrates that many grid cells contain only a single datapoint, whereas some (mostly North Pacific and Arctic) contain over 100 observations. Therefore, while using the raw data would increase the total number of observations to draw from in model training, most would be taken from very similar locations. Gridding the data and taking random draws from the populated grid cells ensures a more even spread of data.

We have now justified this approach in the Methods by stating:

Line 473: This step was necessary to minimize the impact of a few high resolution cruise tracks, which contribute orders of magnitude more datapoints than others.

Line 77 - I assume one of the references here if meant to be for a paper with some of the same authors on sparse marine data (not their one on chemistry integration cited)?

This one:

Keller, C. A. & Evans, M. J. Application of random forest regression to the calculation of gas-phase chemistry within the GEOS-Chem chemistry model v10. *Geosci. Model Dev.* 12, 1209-1225, doi:10.5194/gmd-12-1209-2019 (2019).

Should be:

Sherwen, T., Chance, R. J., Tinel, L., Ellis, D., Evans, M. J., and Carpenter, L. J.: A machine learning based global sea-surface iodide distribution, *Earth Syst. Sci. Data Discuss.*, <https://doi.org/10.5194/essd-2019-40>, in review, 2019.

We thank the reviewer for pointing out this mistake and have updated the reference.

Line 126 - Please see the earlier comment. Although the methods say the effect of the mapping of point data to a coarser map was considered, I am not sure whether the effect of the input variables (features) being gridded to 0.25x0.25 was? If high-resolution variable data were considered (e.g. chlorophyll etc are available to 4km) then this statement would be more persuasive.

See response to the earlier question about grid resolution.

Line 185 - subscript missing for delta methane

We have updated this error.

Line 194 - Please see the earlier comment. Why not just use the inputs for the algorithm (phytoplankton carbon and chlorophyll) instead of providing a product derived from them? Should this just lead to including errors from the other approach? Shouldn't the algorithms used here be able to pick this the parts of the inputs for the NPP prediction that are important for delta methane?

See response to the earlier question about these predictor variables.

Line 203-207 - The exploration of contributions here is interesting. However, as the RF used is interoperable could not more information about the drivers be extracted directly? The use of multiple linear regression here does not seem like the most comprehensible choice for the reader.

We agree that exploring the controls on ΔCH_4 is interesting, and are exploring this more fully through a combination of statistical and mechanistic modeling in the next stage of our project. As discussed in the General Comment above, the focus here is squarely on providing a more robust constraint on marine methane emissions and this is already a lot of ground to cover in a single manuscript. We have therefore attempted to keep the ΔCH_4 interpretation here simple, and focused on explaining the first-order pattern in open-ocean ΔCH_4 in our ensemble-mean distribution. Because this ensemble is a blend of 100,000 different estimates each from both ANN and RRF methods, the direct methods the reviewer mentions cannot be applied. We agree that if we focused on the RRF method alone then more complex diagnostics could be applied, but this would warrant a more detailed treatment than can be afforded in this manuscript.

In presenting our work to the marine methane community (e.g. At the OCB Trace Gas Workshop, Oct. 2018), we have found the simple linear regression analysis has been well received, especially by observationalists who are unfamiliar with machine learning models. Those colleagues have found it intuitive to simply ask which of the predictor variables the ensemble-mean ΔCH_4 pattern is “most similar to”, and also appreciated that this similarity can be confirmed in the original dataset itself. We also note to the reviewer that we have further focused this section on the first-order patterns by removing the discussion of secondary effects identified by multiple linear regression. In response to points raised by Reviewer #2, we felt that it was unfair to use the marginal improvements of the MLR model to favor one potential methanogenesis pathway over another, and we prefer to remain agnostic over the production mechanism.

Line 375 - Please state the number of observations that were binned into the coarser mapped product.

We have added this information to the Methods section:

Line 385 - Why were only 100 ANN and RRF shown in fig 2c? Aesthetic reasons? Were they just choose as random sample? Please state in the caption.

Yes, we chose a random sample for aesthetic reasons. When all 100,000 estimates are plotted it is not possible to distinguish individual datapoints. This is now stated in the caption.

Line 440 - Was the temperature etc with each observation also mapped to a coarser (0.25x0.25) resolution?

No - our method is first to calculate ΔCH_4 for each individual datapoint (using temperature, salinity, etc.), and then bin the ΔCH_4 values to the 0.25° grid.

Line 505 - Why was a transform used for the data used for the RF? Surely this is not needed. What impact on prediction is seen if this is not done?

See response to previous question about data preparation. The transform was performed for consistency between methods.

Line 513 - Please state whether any further randomisation was used (e.g. bootstrap aggregation)?

No additional randomization was performed.

Line 569 - Please give approximate grid resolution in brackets for readability.

Information has been added as requested.

Line 638 - Please consider archiving your output data in a permanent data repository and providing a DOI for it here.

As stated in the response to Reviewer #2's similar concern – we had always intended to archive our ΔCH_4 and flux climatologies in an online repository, but were concerned that our results would make their way into other researchers' work before it has passed the high bar of peer review. We therefore adopted placeholder language in the original submission. We have now set up an archive on the Figshare repository, and reserved a DOI that will activate as soon as our paper is accepted. The reviewer can access the archive using the following private link: <https://figshare.com/s/a6a338252c4b2a209c3d>.

Reviewers' comments:

Reviewer #1 (Remarks to the Author):

I have read the revised version of the manuscript, as well as the authors' responses to the reviewer comments. Most, but not all, of my suggestions and criticisms have been addressed. However, one major issue remains. It is the conclusion that ΔCH_4 is coupled to NPP via aerobic oxidation of methylphosphonate. While the source of methane in the open sea is very likely to be due to methylphosphonate cycling, the control variable is much more likely to be phosphate concentration, as first suggested in Karl et al. (2008; in ref. list), than NPP. This is an emergent field of study and, since the authors first submitted their manuscript, a paper has appeared by Sosa et al. 2019, *Environmental Microbiology* 21: 2402-2414, "Phosphate-limited ocean regions select for bacterial populations enriched in the carbon-phosphorus lyase pathway for phosphonate degradation." Furthermore, long-term studies of ΔCH_4 conducted at Station ALOHA (Wilson et al. 2017, *Geophysical Research Letters*) show a large subdecadal shift from positive ΔCH_4 to negative ΔCH_4 for an environment where NPP is nearly constant but where surface phosphate has changed over time. I realize that there is great value in the satellite-derived ocean color estimate of NPP due to global coverage, but I doubt that NPP, by itself, is the variable that is most responsible for the observations presented. As long as it is presented as a hypothesis (or better yet, a null hypothesis that can be tested), I guess that is fine, but there are many other things that are also correlated to NPP.

The paper is well written and presented and will be a great addition to the growing scientific literature on oceanic methane sources/sinks once it is published.

Reviewer #2 (Remarks to the Author):

"Nature Communications" aims at reporting striking advances in scientific knowledge. The authors report a climatology of ΔCH_4 in the ocean based on interpolation of a global data-base of CH_4 concentration using advanced statistical models, from which the CH_4 emissions are re-estimated. So they meet in part the scope of "Nature Communications" but I think they can still do a better job at explaining and discussing the drivers of the patterns of open oceanic CH_4 ; they totally elude discussing the drivers of CH_4 in the coastal ocean, which could be radically different from those in the open ocean.

Firstly, the relationship of CH_4 and NPP still leaves open the possibility that CH_4 could be related to production in aggregates and/or to transformations of DMS(P), or even in the guts of zooplankton. There is quite compelling recent evidence that CH_4 production in zooplankton guts occurs in areas such as the Baltic Sea (Schmale et al. 2017; Stawiarski et al. 2019; Wäge et al. 2019). In the previous reply to comments, the authors partly address this, mentioning that the lack of correlation with POC export and with the DMS climatology shows that the first two hypothesis cannot be validated with the present analysis. This is an extremely important information for the community and in my opinion needs to be included in the final version of the paper. Regarding zooplankton, the present analysis does not allow to disprove this hypothesis, and this also needs to be mentioned as a possible explanation for the CH_4 -NPP relationship. Indeed, zooplankton biomass is associated to phytoplankton biomass specially if looking at large spatial scales, so a positive CH_4 -NPP relationship is also consistent with the hypothesis of CH_4 production in zooplankton guts. All of this can be mentioned in one or two sentences, so should not increase significantly the length of the present version of the paper (nor compromise future more in depth analysis and papers). By the way, there is no strict length limit for papers in "Nature Communications", so length of text and brevity are not sufficient arguments to decline to address reviewer's suggestions.

Secondly, as mentioned in my previous review, there's no discussion at all on the drivers of CH_4 in the

coastal oceans, when in fact these regions contribute the vast majority of the oceanic CH₄ emissions. Again, this can be addressed in a brief manner, without compromising future analysis and manuscripts (as mentioned in the replies of the authors). For instance possible correlations between CH₄ and NPP or depth should be attempted and reported, looking at least at raw data, or by looking separately at the two model outputs, even if they give different results in coastal waters (note that in open oceanic waters, there is also a strong divergence of the two model outputs in the North Atlantic Ocean polewards of 45°N, which did not impede the analysis of open oceanic CH₄ drivers).

Please note that the occurrence of CH₄ by several groups of phytoplankton has been recently reported by Klintzsch et al. 2019 (extending the previous study of Lenhart on coccolithophores):

Refs

Klintzsch T et al. 2019 Methane production by three widespread marine phytoplankton species: release rates, precursor compounds, and relevance for the environment, <https://www.biogeosciences-discuss.net/bg-2019-245/>

Schmale O et al. 2017, The contribution of zooplankton to methane supersaturation in the oxygenated upper waters of the central Baltic Sea, <https://doi.org/10.1002/lno.10640>

Stawiarski, B., Otto, S., Thiel, V., Gräwe, U., Loick-Wilde, N., Wittenborn, A. K., Schloemer, S., Wäge, J., Rehder, G., Labrenz, M., Wasmund, N., and Schmale, O.: Controls on zooplankton methane production in the central Baltic Sea, *Biogeosciences*, 16, 1-16, <https://doi.org/10.5194/bg-16-1-2019>, 2019.

Wäge J, J F H Strassert, A Landsberger, N Loick-Wilde, O Schmale, B Stawiarski, B Kreikemeyer, G Michel, M Labrenz, 2019: Microcapillary sampling of Baltic Sea copepod gut microbiomes indicates high variability among individuals and the potential for methane production, *FEMS Microbiology Ecology*, Volume 95, Issue 4, fiz024, <https://doi.org/10.1093/femsec/fiz024>

Alberto Borges
University of Liège (Belgium)

Reviewer #3 (Remarks to the Author):

The reviewers have considered all of the points I have raised and more than adequately responded. I support the publication of the manuscript without further revision.

General Comment to all Reviewers

We thank Reviewers #1 and #2 for providing a second round of constructive comments on our manuscript. Both reviewers felt that our discussion of methane production mechanisms in the open ocean remained somewhat incomplete. We have now greatly expanded this section of the manuscript and attempted to provide a balanced and detailed treatment of the various hypotheses. We have now explicitly cited five different potential production mechanisms, and laid out any evidence from our study for and against each. We find that three of these methanogenesis mechanisms are plausible: direct production by phytoplankton, production in zooplankton digestive tracts, and production during methylphosphonate degradation. We note that different pathways may dominate in different regions of the open ocean, and encourage further work to quantify their contributions in our “Outlook” section. Furthermore, in response to the request from Reviewer #2, we have broadened our statistical analysis to include coastal oceans, finding that only seafloor depth is a strong predictor of ΔCH_4 in these regions. The depth vs ΔCH_4 relationship is now illustrated in two new panels added to Fig. 6.

After this second round of revisions, we feel that our manuscript is more complete and much improved, and we are grateful for the input from both reviewers.

Response to Reviewer #1 (Reviewer comments in blue)

I have read the revised version of the manuscript, as well as the authors’ responses to the reviewer comments. Most, but not all, of my suggestions and criticisms have been addressed.

We are pleased that the reviewer felt most of their concerns had been addressed in our previous revisions.

However, one major issue remains. It is the conclusion that ΔCH_4 is coupled to NPP via aerobic oxidation of methylphosphonate. While the source of methane in the open sea is very likely to be due to methylphosphonate cycling, the control variable is much more likely to be phosphate concentration, as first suggested in Karl et al. (2008; in ref. list), than NPP.

We agree that methylphosphonate (MPn) cycling is likely a key CH_4 production mechanism in the open ocean. In response to this reviewer’s comment and those of Reviewer #2, we have provided expanded our discussion of methane production mechanisms, laying out the evidence from our study for and against each. Regarding the MPn mechanism we have reintroduced some text from the original submission of our paper, noting that a multiple linear regression model incorporating NPP and PO_4 explains the mapped ΔCH_4 significantly better than NPP alone, which is consistent with the idea of enhanced MPn cycling under P limitation:

Line 239: We find that a multiple linear regression model combining a positive relationship to NPP and a negative relationship to $[\text{PO}_4]$ explains surface ΔCH_4 significantly better than NPP alone ($\Delta\text{CH}_4 = 5 \times 10^{-3} \text{NPP} - 0.15[\text{PO}_4] - 0.032$, $R^2=0.35$). This relationship is consistent with timeseries evidence for coincident variations

in ΔCH_4 and $[\text{PO}_4]$ in the North Pacific Ocean while NPP remained constant⁴⁸, and supports an important role for MPn cycling as a CH_4 source.

This is an emergent field of study and, since the authors first submitted their manuscript, a paper has appeared by Sosa et al. 2019, *Environmental Microbiology* 21: “Phosphate-limited ocean regions select for bacterial populations enriched in the carbon-phosphorus lyase pathway for phosphonate degradation.” Furthermore, long-term studies of ΔCH_4 conducted at Station ALOHA (Wilson et al. 2017, *Geophysical Research Letters*) show a large subdecadal shift from positive ΔCH_4 to negative ΔCH_4 for an environment where NPP is nearly constant but where surface phosphate has changed over time.

We thank the reviewer for pointing us towards these recent papers. They are both now cited in our manuscript: Wilson et al. (2017) in the statement pasted above (Line 239) and Sosa et al. (2019) in the paragraph pasted in response to the next point (Line 245).

I realize that there is great value in the satellite-derived ocean color estimate of NPP due to global coverage, but I doubt that NPP, by itself, is the variable that is most responsible for the observations presented. As long as it is presented as a hypothesis (or better yet, a null hypothesis that can be tested), I guess that is fine, but there are many other things that are also correlated to NPP.

As outlined above, we now point out that a multiple linear regression model incorporating $[\text{PO}_4]$ gives a better approximation of the ΔCH_4 distribution than NPP alone, supporting the role of PO_4 as a driving variable. We also point out that a combination of methanogenesis pathways may operate in the open ocean, with MPn cycling dominating only in oligotrophic regions:

Line 245: Ultimately, a combination of pathways may control the open ocean surface ΔCH_4 distribution and contribute to its correlation with NPP. Methanogenesis by phytoplankton and in zooplankton guts may dominate in productive ocean regions, with MPn cycling dominating in oligotrophic regions where PO_4 stress acts as the driving variable by selecting for phosphonate decomposing metabolisms⁴⁹.

The paper is well written and presented and will be a great addition to the growing scientific literature on oceanic methane sources/sinks once it is published.

We thank the reviewer for their constructive comments and are pleased that they find our manuscript to be an important contribution to the literature.

Response to Reviewer #2 (Reviewer comments in blue)

“Nature Communications” aims at reporting striking advances in scientific knowledge. The authors report a climatology of ΔCH_4 in the ocean based on interpolation of a global database of CH_4 concentration using advanced statistical models, from which the CH_4 emissions are re-estimated. So they meet in part the scope of “Nature Communications” but I think they can

still do a better job at explaining and discussing the drivers of the patterns of open oceanic CH₄; they totally elude discussing the drivers of CH₄ in the coastal ocean, which could be radically different from those in the open ocean.

We are sorry that the reviewer was not satisfied with the last round of revisions – we strived to address a wide array of issues raised by all three reviewers, but agree that there was still room to be more thorough in discussing the ΔCH_4 vs. NPP relationship, and the drivers of ΔCH_4 variability in the coastal ocean. We are grateful for these suggestions and hope that the revisions we have made in this resubmission do better justice to these important topics.

Firstly, the relationship of CH₄ and NPP still leaves open the possibility that CH₄ could be related to production in aggregates and/or to transformations of DMS(P), or even in the guts of zooplankton. There is quite compelling recent evidence that CH₄ production in zooplankton guts occurs in areas such as the Baltic Sea (Schmale et al. 2017; Stawiarski et al. 2019; Wäge et al. 2019). In the previous reply to comments, the authors partly address this, mentioning that the lack of correlation with POC export and with the DMS climatology shows that the first two hypotheses cannot be validated with the present analysis. This is an extremely important information for the community and in my opinion needs to be included in the final version of the paper. Regarding zooplankton, the present analysis does not allow to disprove this hypothesis, and this also needs to be mentioned as a possible explanation for the CH₄-NPP relationship. Indeed, zooplankton biomass is associated to phytoplankton biomass specially if looking at large spatial scales, so a positive CH₄-NPP relationship is also consistent with the hypothesis of CH₄ production in zooplankton guts. All of this can be mentioned in one or two sentences, so should not increase significantly the length of the present version of the paper (nor compromise future more in depth analysis and papers).

As discussed in the general comment, we have taken a much broader and more balanced approach to the discussion of open ocean methane production in the revised manuscript. We cite five different potential production mechanisms and weigh the evidence for and against each from our study. Regarding the four mechanisms brought to our attention by this reviewer, we state that weak correlations with DMS and POC flux argue against production in sinking aggregates and during DMS transformations, whereas the strong correlation with NPP supports both direct production by phytoplankton or in zooplankton digestive tracts:

Line 222: Methane production has been reported during growth of coccolithophores¹³ and other ubiquitous members of the prymnesiophyte class of marine phytoplankton⁴³, which may contribute in part to the correlation we find between ΔCH_4 and NPP. However, a number of alternative pathways have been proposed for methanogenesis in surface ocean waters, which could give rise to the relationship indirectly. CH₄ may be released from sinking organic aggregates (“marine snow”) that harbor anoxic microzones suitable for methanogenesis⁴⁴, but this should result in a stronger relationship of ΔCH_4 to particulate organic carbon (POC) flux than to NPP, which is not borne out in our analysis ($R^2=0.14$, Table S2). Similarly, CH₄ may be produced in the anoxic digestive tracts of zooplankton and egested to the watercolumn at potentially significant rates¹⁴. Because zooplankton biomass and productivity scales with NPP⁴⁵, this mechanism is broadly consistent with the surface distribution of ΔCH_4 .

In addition, two aerobic pathways have been identified for methanogenesis during the microbial cycling of dissolved organic matter (DOM) compounds, which are ultimately a product of phytoplankton growth (i.e. NPP). First, microbial transformations of dimethylsulfide (DMS, itself produced from DOM precursors) are thought to yield CH₄⁴⁶, but we find only a weak correlation between DMS and ΔCH₄ (Table S2), suggesting this is not an important pathway at the global scale.

Secondly, as mentioned in my previous review, there's no discussion at all on the drivers of CH₄ in the coastal oceans, when in fact these regions contribute the vast majority of the oceanic CH₄ emissions. Again, this can be addressed in a brief manner, without compromising future analysis and manuscripts (as mentioned in the replies of the authors). For instance possible correlations between CH₄ and NPP or depth should be attempted and reported, looking at least at raw data, or by looking separately at the two model outputs, even if they give different results in coastal waters (note that in open oceanic waters, there is also a strong divergence of the two model outputs in the North Atlantic Ocean polewards of 45°N, which did not impede the analysis of open oceanic CH₄ drivers).

We have now broadened our statistical analyses to consider coastal regions (<2000m deep) in addition to open ocean regions, and have therefore changed the subtitle of this section in our manuscript to "Drivers of ocean methane disequilibrium". Table S2 has been expanded to show correlation coefficients between log₁₀(ΔCH₄) and predictor variables in the coastal ocean, demonstrating that the only relationship is against seafloor depth (even more so against log₁₀ depth). We illustrate this relationship for both mapped and observed ΔCH₄ on log-log correlation plots which have been added to Fig. 6, thus giving equal weight to the coastal ocean and open ocean in this Figure. We have also provided an expanded discussion of the factors that likely drive this strong relationship between ΔCH₄ and depth. The section dedicated to the coastal ocean now reads:

Line 193: In coastal ocean regions (<2000m) where ΔCH₄ spans orders of magnitude, log₁₀(ΔCH₄) only correlates strongly with seafloor depth (R²=0.37) whereas other predictor variables can explain at most ~10% of its spatial variance (Table S2). The correlation is further strengthened against log-transformed depth (R²=0.55), indicating that the first order pattern inferred by our machine learning models is a decline in ΔCH₄ away from coastlines following a power law relationship against seafloor depth (z_{sf}): $\Delta\text{CH}_4 = 67z_{sf}^{-0.7}$. A very similar relationship can be derived directly from the raw dataset used to train our models (Fig. 6b), and the same qualitative pattern is apparent in observations across the shelf at individual locations²². The strong dependence of ΔCH₄ on depth reflects the important role of the seafloor as a CH₄ source to the surface ocean, either supplied by rising gas bubbles that dissolve within tens of meters of the seafloor (Fig. 5a), or by diffusion from anoxic sediments and transport to the surface. In the latter case, bathymetry controls both the rain rate of organic carbon that fuels anaerobic metabolism in the sediments⁸, and the mixing timescale between bottom waters and the surface mixed layer. The lack of any strong correlation against other predictor variables suggests that the environmental controls of seafloor CH₄ sources are complex and vary significantly between regions.

Please note that the occurrence of CH₄ by several groups of phytoplankton has been recently reported by Klintzsch et al. 2019 (extending the previous study of Lenhart on coccolithophores):

Reviewer #2 continues to point us towards very interesting studies, for which we are grateful!
This one has now been cited in the revised manuscript.

REVIEWERS' COMMENTS:

Reviewer #1 (Remarks to the Author):

none, except 'good job'

Reviewer #2 (Remarks to the Author):

The authors have addressed very satisfactorily my comments from the previous round of review, and I find the present discussion of the CH₄ patterns well balanced. The new version of Figure 6 is magnificent and this will become an instant classic. This figure conveys a very important message, that CH₄ in the open ocean and coastal ocean seem to have different origins.

A very small (final) suggestion would be to check if temperature contributes to the variability of the deltaCH₄ vs depth relation of the coastal zone, with a MLR of deltaCH₄ vs (T;depth) for the raw data, or regressions of deltaCH₄ versus T per bins of depth. Several local studies have shown a positive relation between CH₄ and water temperature, it would be interesting if such pattern emerges in the global data-set.

Alberto Borges, University of Liège.

Response to Reviewer #1 (Reviewer comments in blue)

(Comments to author) none, except 'good job'

Thanks! The reviewer's comments over two rounds of review have greatly benefited our paper.

Response to Reviewer #2 (Reviewer comments in blue)

The authors have addressed very satisfactorily my comments from the previous round of review, and I find the present discussion of the CH₄ patterns well balanced. The new version of Figure 6 is magnificent and this will become an instant classic. This figure conveys a very important message, that CH₄ in the open ocean and coastal ocean seem to have different origins.

We are pleased that the reviewer is fully satisfied by our last round of revisions, and thank them for pushing us to go the extra mile in our paper. We agree that Figure 6 is quite powerful and may well become a mainstay of presentations given about the marine methane cycle.

A very small (final) suggestion would be to check if temperature contributes to the variability of the deltaCH₄ vs depth relation of the coastal zone, with a MLR of deltaCH₄ vs (T,depth) for the raw data, or regressions of deltaCH₄ versus T per bins of depth. Several local studies have shown a positive relation between CH₄ and water temperature, it would be interesting if such pattern emerges in the global data-set.

We thank the reviewer for this final suggestion. As requested, we tested whether a multiple regression model combining $\log_{10}(\text{depth})$ and temperature (T) can explain coastal ΔCH_4 better than depth alone. We found that the temperature effect is extremely small: $\log_{10}(\Delta\text{CH}_4) = 0.7\log_{10}(\text{depth}) + 2.7 \times 10^{-6}T + 1.8$, and that there is no significant improvement over the simpler depth relationship ($R^2=0.5479$ with or without temperature). While we agree that temperature appears important in some local studies, this relationship does not emerge from the global dataset. We have decided not to further complicate our text by discussing this null result, given it is already addressed by the statement:

"The lack of strong relationships to other predictor variables suggests that the environmental controls of seafloor CH₄ sources are complex and vary significantly between regions."